# Physically informed artificial neural networks for atomistic modeling of materials

G.P.Purja Pun[1], R. Batra[2], R. Ramprasad [2] & Y. Mishin[1]

Large-scale atomistic computer simulations of materials heavily rely on interatomic potentials predicting the energy and Newtonian forces on atoms. Traditional interatomic potentials are based on physical intuition but contain few adjustable parameters and are usually not accurate. The emerging machine-learning (ML) potentials achieve highly accurate interpolation within a large DFT database but, being purely mathematical constructions, suffer from poor transferability to unknown structures. We propose a new approach that can drastically improve the transferability of ML potentials by informing them of the physical nature of interatomic bonding. This is achieved by combining a rather general physics-based model (analytical bond-order potential) with a neural-network regression. This approach, called the physically informed neural network (PINN) potential, is demonstrated by developing a general-purpose PINN potential for Al. We suggest that the development of physics-based ML potentials is the most effective way forward in the field of atomistic simulations.

---

[1] Department of Physics and Astronomy, MSN 3F3, George Mason University, Fairfax, VA 22030, USA. [2] School of Materials Science and Engineering, Georgia Institute of Technology, Atlanta, GA 30332, USA. Correspondence and requests for materials should be addressed to Y.M. (email: ymishin@gmu.edu)

Large-scale molecular dynamics (MD) and Monte Carlo (MC) simulations of materials are traditionally implemented using classical interatomic potentials predicting the potential energy and Newtonian forces acting on atoms. Computations with such potentials are very fast and afford access to systems with millions of atoms and MD simulation times up to hundreds of nanoseconds. Such simulations span a wide range of time and length scales and constitute a critical component of the multiscale approach in materials modeling and computational design.

Several functional forms of interatomic potentials have been developed over the years, including the embedded-atom method (EAM)[1–3], the modified EAM (MEAM)[4], the angular-dependent potentials[5], the charge-optimized many-body potentials[6], reactive bond-order potentials[7–9], and reactive force fields[10] to name a few. These potentials address particular classes of materials or particular types of applications. Their functional forms depend on the physical and chemical models chosen to describe interatomic bonding in the respective class of materials.

A common feature of all traditional potentials is that they express the potential energy surface (PES) of the system, $E = E(\mathbf{r}_1, ..., \mathbf{r}_N, \mathbf{p})$, as a relatively simple function of atomic coordinates ($\mathbf{r}_1, ..., \mathbf{r}_N$), $N$ being the number of atoms (Fig. 1a). Knowing the PES, the forces acting on the atoms can be computed by differentiation and used in MD simulations. The potential functions depend on a relatively small number of fitting parameters $\mathbf{p} = (p_1, ..., p_m)$ (typically, $m = 10–20$) and are optimized (trained) on a relatively small database of experimental data and first-principles density functional theory (DFT) calculations. The

traditional potentials are, of course, much less accurate than DFT calculations. Nevertheless, many of them demonstrate a reasonably good transferability to atomic configurations lying well outside the training dataset. This important feature owes its origin to the incorporation of at least some basic physics in the potential form. As long as the nature of chemical bonding remains the same as assumed during the potential development, the potential can predict the system energy adequately even for new configurations not seen during the training process. Unfortunately, the construction of good quality potentials is a long and painful process requiring personal experience and intuition and is more art than science[8,11]. In addition, the traditional potentials are specific to a particular class of materials and cannot be easily extended to other materials or improved in a systematic manner.

During the past decade, a new direction has emerged wherein interatomic potentials are developed by employing machine-learning (ML) methods[12–22]. The idea was originally conceived in the chemistry community in the 1990s in the effort to improve the accuracy of inter-molecular force fields[23,24], an approach that was later adopted by the physics and materials science communities. The general idea is to forego the physical insights and reproduce the PES by interpolating between DFT data points using high-dimensional nonlinear regression methods such as the Gaussian process regression[19,25–27], interpolating moving least squares[28], kernel ridge regression[12,20,21], compressed sensing[29,30], gradient-domain machine-learning model[31], or the artificial neural network (NN) approach[13–18,32–38]. If properly trained, a ML potential can predict the system energy with a nearly DFT accuracy (a few meV/atom). ML potentials are not specific to a particular class of materials or type of chemical bonding. They can be improved systematically if weaknesses are discovered or new DFT data become available. The training process can be implemented on-the-fly by running ab initio MD simulations[26].

A major weakness of ML potentials is their poor transferability. Being purely mathematical constructions devoid of any physical meaning, they can accurately interpolate the energy between the training configurations but are generally incapable of properly extrapolating the energy to unknown atomic environments. As a result, the performance of ML potentials outside the training domain can be very poor. There is no reason why a purely mathematical extrapolation scheme would deliver physically meaningful results outside the training database. This explains why the existing ML potentials are usually (with rare exceptions[39]) narrowly focused on, and only tested for, a particular type of physical properties. This distinguishes them from the traditional potentials which, although less accurate, are designed for a much wider range of applications and diverse properties.

In this work we propose a new approach that can drastically improve the transferability of ML potentials by informing them of the physical nature of interatomic bonding. We focus on NN potentials as an example, but the approach is general and can be readily extended to other methods of nonlinear regression. Like all ML potentials, the proposed physically informed NN (PINN) potentials are trained using a large DFT dataset. However, by contrast to the existing, mathematical NN potentials, the PINN potentials incorporate the basic physics and chemistry of atomic interactions leveraged by the extraordinary adaptivity and trainability of NNs. The PINN potentials thus strike a golden compromise between the two extremes represented by the traditional, physics-guided interatomic potentials, and the mathematical NN potentials.

The general idea of combining traditional interatomic potentials with NNs was previously discussed by Malshe et al.[40], who constructed an adjustable Tersoff potential[41–43] for a $Si_5$ cluster. Other authors have also applied machine-learning methods to parameterize physics-based models of molecular interactions,

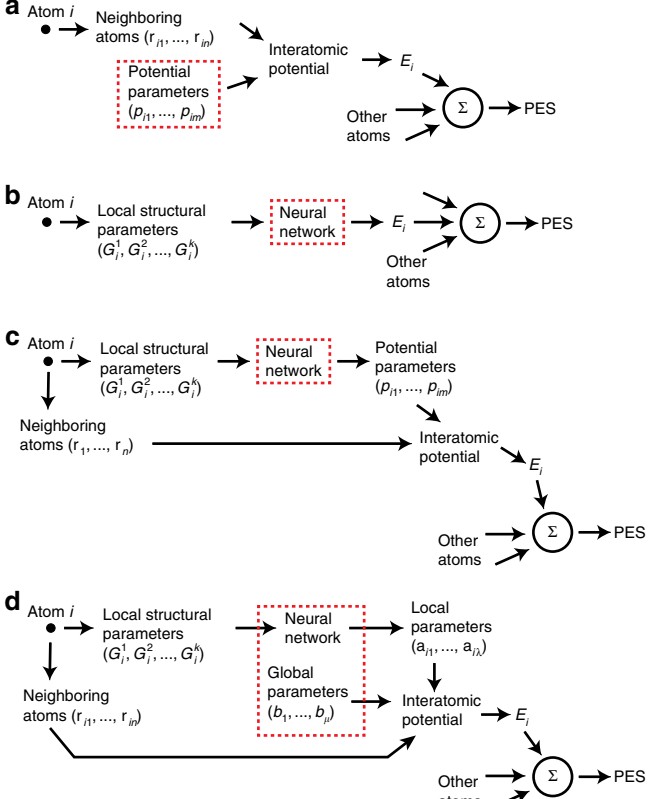

**Fig. 1** Flowcharts of the development of atomistic potentials. **a** Traditional interatomic potential. **b** Mathematical NN potential. **c** Physically informed NN (PINN) potential with all-local parameters. **d** PINN potential with parameters divided into local and global. The dashed rectangle outlines the objects requiring parameter optimization. PES is the potential energy surface of the material

primarily in the context of broad exploration of the compositional space of molecular (mostly organic) matter[44–46]. Glielmo et al.[47] recently proposed to construct $n$-body Gaussian process kernels to capture the $n$-body nature of atomic interactions in physical systems. The PINN potentials proposed in this paper are inspired by such approaches but extend them to (1) more advanced physical models with a broad applicability, and (2) large-scale systems by introducing local energies $E_i$ linked to local structural parameters $G_i^l$. The focus is placed on the exploration of the configurational space of defected solids and liquids in single-component and, in the future, binary or multicomponent systems. The main goal is to improve the transferability of interatomic potentials to unknown atomic environments while keeping the same level of accuracy of training as normally achieved with mathematical machine-learning potentials.

## Results

**Physically informed neural network potentials**. The currently existing, mathematical NN potentials[13–18,32–36] partition the total energy $E$ into a sum of atomic energies, $E = \sum_i E_i$. A single NN is constructed to express each atomic energy $E_i$ as a function of a set of local fingerprint parameters (also called symmetry parameters[13]) $(G_i^1, G_i^2, ..., G_i^k)$. These parameters encode the local environments of the atoms. The network is trained by minimizing the error between the energies predicted by the NN and the respective DFT total energies for a large set of atomic configurations. The flowchart of the method is depicted in Fig. 1b.

The proposed PINN model is based on the following considerations. A traditional, physics-based potential can always be trained to reproduce the energy of any given atomic configuration with any desired accuracy. Of course, this potential will not work well for other configurations. Imagine, however, that the potential parameters have been trained for a large set of reference structures, one structure at a time, each time producing a different parameter set $\mathbf{p}$. Suppose that, during the subsequent simulations, we have a way of identifying, on the fly, a reference structure closest to any current atomic configuration. Then the accuracy of the simulation can be drastically improved by dynamically choosing the best set of potential parameters for every atomic configuration accoutered during the simulation. Now, since the atomic energy $E_i$ only depends on the local environment of atom $i$, the best parameter set for computing $E_i$ can be chosen by only examining the local environment of this atom. The energies of different atoms are then computed by using different, environment-dependent, parameter sets while keeping the same, physics-motivated functional form of the potential.

Instead of generating and storing a large set of discrete reference structures, we can construct a continuous NN-based function mapping the local environment of every atom on a parameter set of the interatomic potential optimized for that particular environment. Specifically, the local structural parameters (fingerprints) $G_i^l$ ($l = 1, ..., k$) of every atom $i$ are fed into the network, which then maps them to the optimized parameter set $\mathbf{p}_i$ appropriate for atom $i$. Mathematically, the local energy takes the functional form

$$E_i = E_i\big(\mathbf{r}_{i1}, ..., \mathbf{r}_{in}, \mathbf{p}_i\big(G_i^l(\mathbf{r}_{i1}, ..., \mathbf{r}_{in})\big)\big), \qquad (1)$$

where $(\mathbf{r}_{i1}, ..., \mathbf{r}_{in})$ are atomic positions in the vicinity of atom $i$.

In comparison with the direct mapping $G_i^l \mapsto E_i$ implemented by the mathematical NN potentials, we have added an intermediate step: $G_i^l \mapsto \mathbf{p}_i \mapsto E_i$. The first step is executed by the NN and the second by a physics-based interatomic potential. A flowchart of the two-step mapping is shown in Fig. 1c. It is important to emphasize that this intermediate step does not degrade the accuracy relative to the direct mapping, because a

feedforward NN can always be trained to execute any real-valued function[48,49]. Thus, for any functional form of the potential, the NN can always adjust its architecture, weights and biases to achieve the same mapping as in the direct method. However, since the chosen potential form captures the essential physics of atomic interactions, the proposed PINN potential will display a better transferability to new atomic environments. Even if the potential parameters predicted by the NN for an unknown environment are not very accurate, the physics-motivated functional form will ensure that the results remain at least physically meaningful. This physics-guided extrapolation is likely to be more reliable than the purely mathematical extrapolation inherent in the existing NN potentials. Obviously, the same reasoning applies to the interpolation process as well, which can also be more accurate.

The functional form of the PINN potential must be general enough to be applicable across different classes of materials. In this paper we chose a simple analytical bond-order potential (BOP)[50–52] that must work equally well for both covalent and metallic materials. For a single-component system, the BOP functions are specified in the Methods section. They capture the physical and chemical effects such as the pairwise repulsion between atoms, the angular dependence of the chemical bond strength, the bond-order effect (the more neighbors, the weaker the bond), and the screening of chemical bonds by surrounding atoms. In addition to being appropriate for covalent bonding, the proposed BOP form reduces to the EAM formalism in the limit of metallic bonding.

**Example: PINN potential for Al**. To demonstrate the PINN method, we have constructed a general-purpose potential for aluminum. The training and validation datasets were randomly selected from a pre-existing DFT database[20,21]. Some additional DFT calculations have also been performed using the same methodology as in refs. [20,21]. The selected DFT supercells represent seven crystal structures for a large set of atomic volumes under isotropic tension and compression, several slabs with different surface orientations, including surfaces with adatoms, a supercell with a single vacancy, five different symmetrical tilt grain boundaries, and an unrelaxed intrinsic stacking fault on the (111) plane with different translational states along the [211] direction. The database also includes several isolated clusters with the number of atoms ranging from 2 (dimer) to 79. The ground-state face centered cubic (FCC) structure was additionally subject to uniaxial tension and compression in the [100] and [111] directions at 0 K temperature. Most of the atomic configurations were snapshots of DFT MD simulations in the microcanonical (NVE) or canonical (NVT or NPT) ensembles for several atomic volumes at several temperatures. Some of the high-temperature configurations were part-liquid, part crystalline. In total, the database contains 3649 supercells (127592 atoms). More detailed information about the database can be found in the Supplementary Tables 1 and 2. To avoid overfitting or selection bias, the 10-fold cross-validation method was used during the training. The database was randomly partitioned in 10 subsets. One of them was set aside for validation and the remaining data was used for training. The process repeated 10 times for different choices of the validation subset.

The local structural parameters $G_i^l$ chosen for Al are specified in the Methods section. The NN contained two hidden layers with the same number of nodes in each. This number was increased until the training process produced a PINN potential with the root-mean-square error (RMSE) of training and validation close to 3–4 meV per atom, which was set as our goal. This is the level of accuracy of the DFT energies included in the

database. For comparison, a mathematical NN potential was constructed using the same methodology. The number of hidden nodes of the NN was adjusted to give about the same number of fitted parameters and to achieve approximately the same RMSE of training and validation as for the PINN potential. Table 1 summarizes the training and validation errors averaged over the 10 cross-validation runs. One PINN and one NN potential were selected for a more detailed examination reported below.

Figure 2 and Supplementary Fig. 1 demonstrate excellent correlation between the predicted and DFT energies over a 7 eV per atom wide energy range for both potentials. The error distribution has a near-Gaussian shape centered at zero. Examination of errors in individual groups of structures (Supplementary Fig. 2) shows that the largest errors originate from the crystal structures (especially FCC, HCP, and simple hexagonal) subjected to large expansion.

Table 2 summarizes some of the physical properties of Al predicted by the potentials in comparison with DFT data from the literature. There was no direct fit to any of these properties, although atomic configurations most relevant to some of the properties were represented in the training dataset. While both potentials agree with the DFT data well, the PINN potential tends

to be more accurate for most properties. For the [110] self-interstitial dumbbell, the NN potential predicts an unstable configuration that spontaneously rotates to the [100] orientation, whereas the PINN potential correctly predicts such configurations to be metastable. Figure 3 shows the linear thermal expansion factor as a function of temperature predicted by the potentials in comparison with experimental data. The PINN potential displays good agreement with experiment without direct fit, whereas the NN potential overestimates the thermal expansion at high temperatures. (The discrepancies at low temperatures are due to the quantum effects that are not captured by classical simulations.) As another test, the radial distribution function and the bond angle distribution in liquid Al were computed at several temperatures for which experimental and/or DFT data are available (Supplementary Figs 4 and 5). In this case, both potentials were found to perform equally well. Any small deviations from the published DFT calculations are within the uncertainty of the different DFT flavors (exchange-correlation functionals).

For testing purposes, we computed the energies of the remaining groups of structures that were part of the original DFT database[20,21] but were not used here for training or validation. The full information about the testing dataset (26,425 supercells containing a total of 2,376,388 atoms) can be found in the Supplementary Table 3. For example, Fig. 4 compares the energies predicted by the potentials with DFT energies from high-temperature MD simulations for a supercell containing an edge dislocation or HCP Al. In both cases, the PINN potential is obviously more accurate. The remaining testing cases are presented in the Supplementary Figs. 6–10. Although there are cases where both potentials perform equally well, in most cases the PINN potential predicts the energies of unknown atomic configurations more accurately than the NN potential.

For further testing, the energies of the crystal structures of Al were computed for atomic volumes both within and beyond the training interval. Both potentials accurately reproduce the DFT energy–volume relations for all volumes spanned by the DFT

| Table 1 Fitting and validation errors of the straight NN and PINN models | | | | |
|---|---|---|---|---|
| Model | NN architecture | Number of parameters | RMSE of training (meV per atom) | RMSE of validation (meV per atom) |
| NN | 60 × 16 × 16 × 1 | 1265 | 3.36 | 3.85 |
| NN′ | 47 × 18 × 18 × 1 | 1225 | 3.62 | 3.54 |
| PINN | 60 × 15 × 15 × 8 | 1283 | 3.46 | 3.59 |

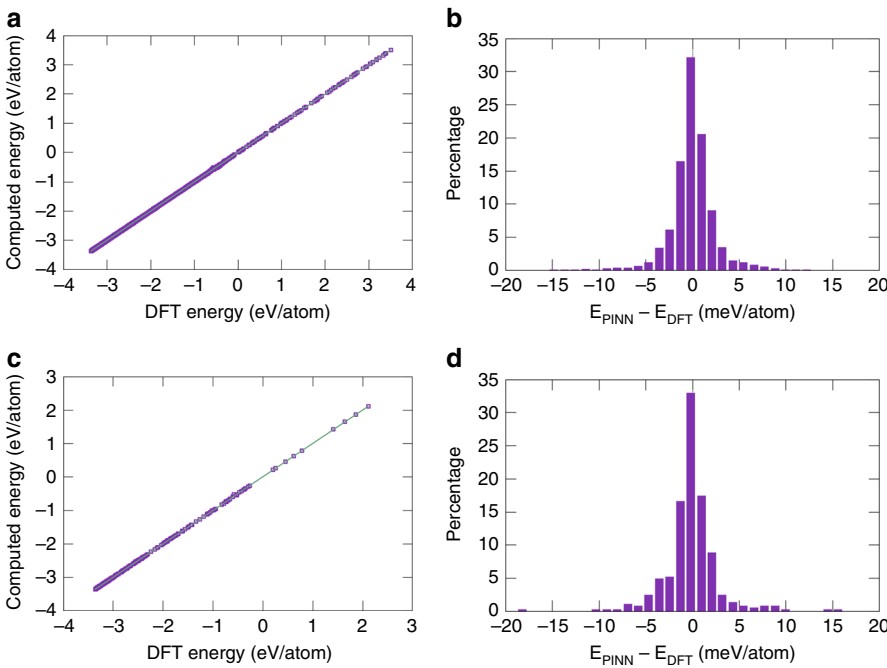

**Fig. 2** Accuracy of the PINN potential. **a**, **c** Energies of atomic configurations in the **a** training and **c** validation datasets computed with the PINN potential versus DFT energies. The straight line represents the perfect fit. **b**, **d** Error distributions in the **b** training and **d** validation datasets

**Table 2 Aluminum properties predicted by the PINN and NN potentials**

| Property | DFT | NN | PINN |
|---|---|---|---|
| $E_0$ (eV per atom) | $-3.7480^a$ | $-3.3606$ | $-3.3609$ |
| $a_0$ (Å) | $4.039^{a,d}$; $3.9725–4.0676^c$ | 4.0409 | 4.0396 |
| $B$ (GPa) | $83^a$; $81^f$ | 80 | 79 |
| $c_{11}$ (GPa) | $104^a$; $103–106^d$ | 108 | 117 |
| $c_{12}$ (GPa) | $73^a$; $57–66^d$ | 66 | 60 |
| $c_{44}$ (GPa) | $32^a$; $28–33^d$ | 25 | 32 |
| $\gamma_s(100)$ (J m$^{-2}$) | $0.92^b$ | 0.897 | 0.899 |
| $\gamma_s(110)$ (J m$^{-2}$) | $0.98^b$ | 0.986 | 0.952 |
| $\gamma_s(111)$ (J m$^{-2}$) | $0.80^b$ | 0.837 | 0.819 |
| $E_v^f$ (eV) | $0.665–1.346^c$; $0.7^e$ | 0.640 | 0.678 |
| $E_v^f$ (eV) unrelaxed | $0.78^e$ | 0.71 | 0.77 |
| $E_v^m$ (eV) | $0.304–0.621^c$ | 0.627 | 0.495 |
| $E_i^f$ ($T_d$) (eV) | $2.200–3.294^c$ | 2.683 | 2.840 |
| $E_i^f$ ($O_h$) (eV) | $2.531–2.948^c$ | 1.600 | 2.367 |
| $E_i^f$ $\langle 100 \rangle$ (eV) | $2.295–2.607^c$ | 1.529 | 2.246 |
| $E_i^f$ $\langle 110 \rangle$ (eV) | $2.543–2.981^c$ | 1.529* | 2.713 |
| $E_i^f$ $\langle 111 \rangle$ (eV) | $2.679–3.182^c$ | 2.631 | 2.815 |
| $\gamma_{SF}$ (mJ m$^{-2}$) | $134^i$; $146^g$; $158^h$ | 128 | 121 |
| $\gamma_{us}$ (mJ m$^{-2}$) | $162^j$; $169^i$; $175^h$ | 143 | 132 |

The potential predictions are compared with DFT calculations from the literature
$E_0$ equilibrium cohesive energy, $a_0$ equilibrium lattice parameter, $B$ bulk modulus, $c_{ij}$ elastic constants, $\gamma_s$ surface energy, $E_v^f$ vacancy formation energy, $E_v^m$ vacancy migration barrier, $E_i^f$ interstitial formation energy for the tetrahedral ($T_d$) and octahedral ($O_h$) positions and split dumbbell configurations with different orientations, $\gamma_{SF}$ intrinsic stacking fault energy, $\gamma_{us}$ unstable stacking fault energy. All defect energies are statically relaxed unless otherwise indicated
$^a$Ref. [61]; $^b$ref. [62]; $^c$ref. [63]; $^d$ref. [64]; $^e$ref. [65]; $^f$ref. [66]; $^g$ref. [67]; $^h$ref. [68]; $^i$ref. [69]; $^j$ref. [70]
*Unstable and flips to the $\langle 100 \rangle$ dumbbell orientation

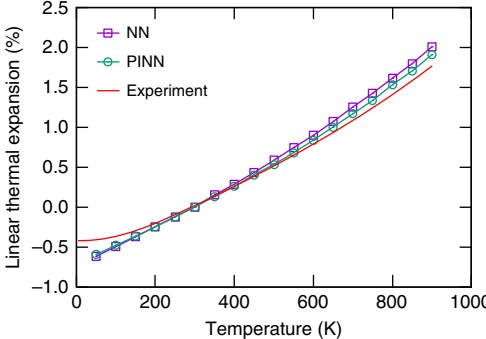

**Fig. 3** Linear thermal expansion of Al relative to room temperature (295 K) predicted by the PINN and NN potentials in comparison with experiment[60]

database (Fig. 5 and Supplementary Fig. 3). However, extrapolation to larger or smaller volumes reveals significant differences. For example, the PINN potential correctly predicts that the crystal energy continues to rapidly increase under strong compression (repulsive interaction mode). In fact, the extrapolated PINN energy goes exactly through the new DFT points that were not included in the training or validation datasets, see examples in Fig. 6. By contrast, the energy predicted by the NN model immediately develops wiggles and strongly deviates from the physically meaningful repulsive behavior. Such artifacts were found for other structures as well.

To demonstrate that the unphysical behavior exhibited by the NN potential is not a specific feature of our structural parameters $G_i^l$ or the training method, we constructed another NN potential using a third-party NN-training package PROPhet[53]. This potential, which we refer to as NN′, uses the Behler-Parrinello symmetry functions[13], which are different from our structural descriptor $G_i^l$. The NN-training algorithm is also different. A $47 \times 18 \times 18 \times 1$ network containing 1225 fitting parameters was trained on exactly the same DFT database to about the same accuracy as the NN and PINN potentials (Table 1). Figure 6 shows that the NN′ potential behaves in a similar manner as our NN potential, closely following the DFT energies within the training/validation domain and becoming unphysical as soon as we step outside this domain.

While the atomic forces were not used for either training or validation, they were compared with the DFT forces once the training was complete. For the validation dataset, this comparison probes the accuracy of interpolation, whereas for the testing dataset the accuracy of extrapolation. As expected, for the validation dataset the PINN forces are in better agreement with DFT calculations than the NN forces (RMSE $\approx 0.1$ eV Å$^{-1}$ versus $\approx 0.2$ eV Å$^{-1}$) as illustrated in Fig. 7a, b. For the testing dataset, the advantage of the PINN model in force predictions is even more significant. For example, for the dislocation and HCP cases discussed above, the PINN potential provides more accurate predictions (RMSE $\approx 0.1$ eV Å$^{-1}$) than the NN potential (RMSE $\approx 0.4$ eV Å$^{-1}$ for the dislocation and 0.6 eV Å$^{-1}$ for the HCP case) (Fig. 7c, f). This advantage persists for all other groups of structures from the testing database.

It was also interesting to compare the PINN potential with traditional, parameter-based potentials for Al. One of them was the widely accepted EAM Al potential[54] that had been fitted to a mix of experimental and DFT data. The other was a BOP potential of the same functional form as in the PINN model. Its parameters were fitted in this work using the same DFT database as for the PINN/NN potentials and then fixed once and for all. Figure 8 compares the DFT energies with the energies predicted by the EAM and BOP models across the entire set of reference configurations. The PINN predictions are shown for comparison. The plots demonstrate that the traditional, fixed-parameter models generally follow the correct trend but become increasingly less accurate as the structures deviate from the equilibrium, low-energy atomic configurations. The adaptivity to the local atomic environments built into the PINN potential greatly improves the accuracy.

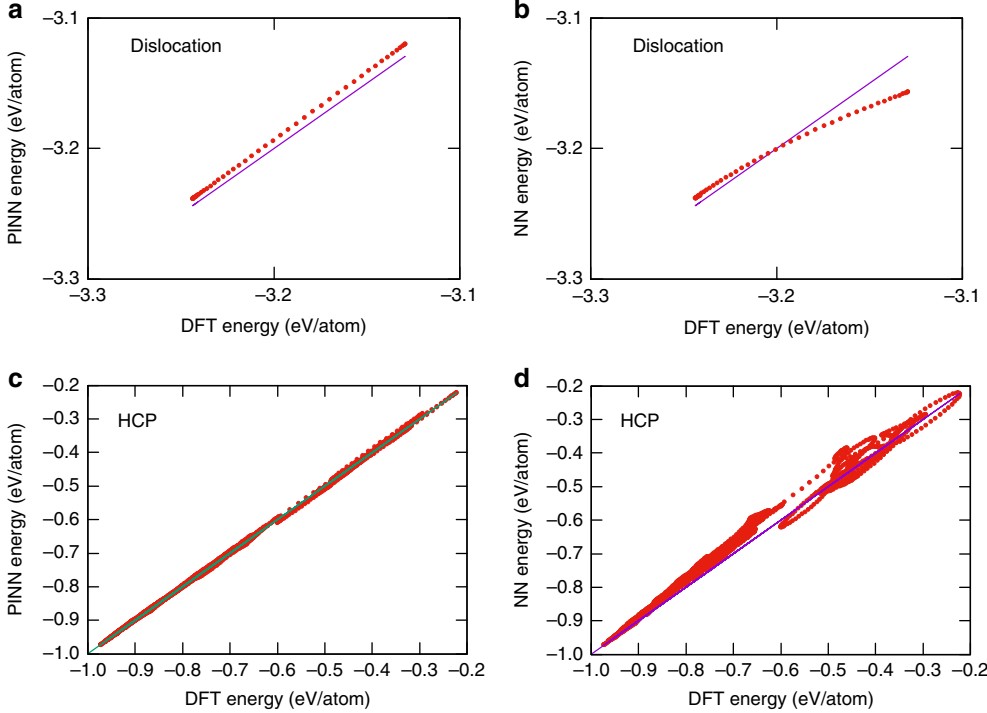

**Fig. 4** Testing of the NN and PINN potentials. **a**, **b** Energy of an edge dislocation in Al in NVE MD simulations starting at 700 K. **c**, **d** Energy of HCP Al in NVT MD simulations at 1000, 1500, 2000, and 4000 K. The energies predicted by the PINN (**a**, **c**) and NN (**b**, **d**) potentials are compared with DFT calculations from[20,21]. The straight lines represent the perfect fit

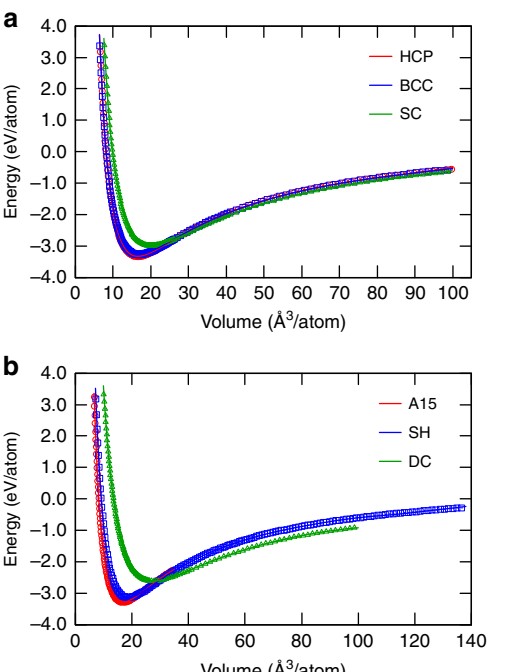

**Fig. 5** Energy–volume relations for Al crystal structures. Comparison of the energies predicted by the PINN potential (lines) and by DFT calculations (points). **a** Hexagonal close-packed (HCP), body-centered cubic (BCC), and simple cubic (SC) structures. **b** A15 (Cr$_3$Si prototype), simple hexagonal (SH), and diamond cubic (DC) structures

## Discussion

The proposed PINN potential model is capable of achieving the same high accuracy in interpolating between DFT energies on the PES as the currently existing mathematical NN potentials. The construction of PINN potentials requires the same type of DFT database, is equally straightforward, and does not heavily rely on human intuition. However, extrapolation outside the domain of atomic configurations represented in the training database is now based on a physical model of interatomic bonding. As a result, the extrapolation becomes more reliable, or at least more failure-proof, than the purely mathematical extrapolation. The accuracy of interpolation can also be improved for the same reason. As an example, the PINN Al potential constructed in this paper demonstrates better accuracy of interpolation and significantly improved transferability than a regular NN potential with about the same number of parameters. The advantage of the PINN potential is especially strong for atomic forces, which are important for molecular dynamics. The potential could be used for accurate simulations of mechanical behavior and other processes in Al. Construction of general-purpose PINN potentials for Si and Ge is currently in progress.

We believe that the development of physics-based ML potentials is the best way forward in this field. Such potentials need not be limited to NNs or the particular BOP model adopted in this paper. Other regression methods can be employed and the interatomic bonding model can be made more sophisticated, or the other way round, simpler in the interest of speed.

Other modifications are envisioned in the future. For example, not all potential parameters are equally sensitive to local environments. To improve the computational efficiency, the parameters can be divided into two subsets[40]: local parameters $\mathbf{a}_i = (a_{i1}, ..., a_{i\lambda})$ adjustable according to the local environments as discussed above, and global parameters $\mathbf{b} = (b_1, ..., b_\mu)$ that are fixed after the optimization and used for all environments (as in the traditional potentials). The potential format now becomes

$$E_i = E_i\big(\mathbf{r}_{i1}, ..., \mathbf{r}_{in}, \mathbf{a}_i\big(G_i^l(\mathbf{r}_{i1}, ..., \mathbf{r}_{in})\big), \mathbf{b}\big). \qquad (2)$$

During the training process, the global parameters $\mathbf{b}$ and the network weights and biases are optimized simultaneously, as

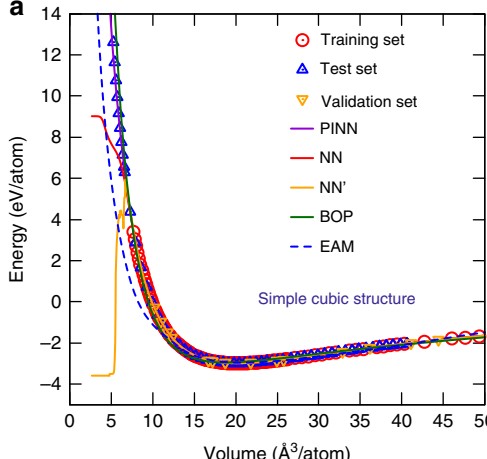

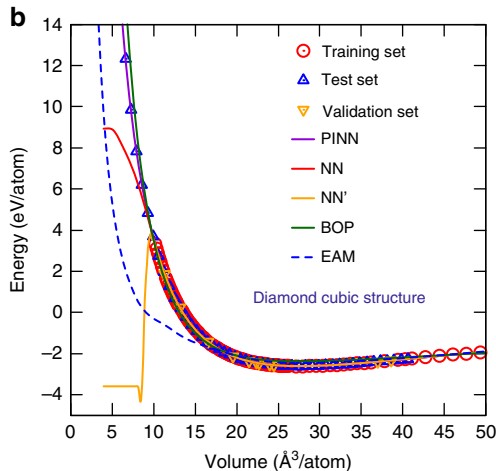

**Fig. 6** Zoom into the repulsive part of the energy–volume relations predicted by the PINN, NN, NN′, EAM, and BOP potentials (curves) and DFT calculations (points)

shown in Fig. 1d. Extension of PINN potentials to binary and multicomponent systems is another major task for the future.

All ML potentials are orders of magnitude faster than straight DFT calculations but inevitably much slower than the traditional potentials. Preliminary tests indicate that PINN potentials are about 25% slower than the regular NN potentials for the same number of parameters, the extra overhead being due to the BOP calculation. However, the computational efficiency depends on the parallelization method and computer architecture. All computations reported in this paper utilized in-house software parallelized with MPI for training and with OpenMP for MD and MC simulations (see example in Supplementary Fig. 14). Collaborative work is underway to develop highly scalable HPC software packages for physically informed ML potential training and MD/MC simulations using multiple CPUs or GPUs, or both. The results will be reported in a forthcoming paper.

## Methods

**Local structural parameters**. There are many possible ways of choosing local structural parameters[13–18,34,36]. After trying several options, the following set of $G_i^{l}$'s was selected. For an atom $i$, we define

$$g_i^{(m)} = \sum_{j,k} P_m\left(\cos\theta_{ijk}\right) f(r_{ij}) f(r_{ik}), \quad m = 0, 1, 2, \dots, \quad (3)$$

where $r_{ij}$ and $r_{ik}$ are distances to atoms $j$ and $k$, respectively, and $\theta_{ijk}$ is the angle between the bonds $ij$ and $ik$. In Eq. (3), $P_m(x)$ is the Legendre polynomial of order

$m$ and

$$f(r) = \frac{1}{\sigma^3} e^{-(r-r_0)^2/\sigma^2} f_c(r) \quad (4)$$

is a truncated Gaussian of width $\sigma$ centered at point $r_0$. The truncation function $f_c(r)$ is defined by

$$f_c(r) = \begin{cases} \frac{(r-r_c)^4}{d^4 + (r-r_c)^4} & r \leq r_c \\ 0, & r \geq r_c. \end{cases} \quad (5)$$

This function and its derivatives up to the third go to zero at a cutoff distance $r_c$. The parameter $d$ controls the truncation range.

For example, $P_0(x) = 1$ and $g_i^{(0)}$ characterizes the local atomic density near atom $i$. Likewise, $P_1(x) = x$ and $g_i^{(1)}$ can be interpreted as the dipole moment of a set of unit charges placed at the atomic positions $j$ and $k$. As such, this parameter measures the degree of local deviation from spherical symmetry in the environment ($g_i^{(1)} = 0$ for spherical symmetry). For $m = 2$, we have $P_2(x) = (3x^2 - 1)/2$ and $g_i^{(2)}$ is related to the quadrupole moment of a set of unit charges placed at the atomic positions around atom $i$. We found that polynomials up to degree $m = 6$ should be included to accurately represent the diverse atomic environment. Each $g_i^{(l)}$ is computed for several values of $\sigma$ and $r_0$ spanning a range of interatomic distances. For each atom, the set of $k$ $g_i^{(m)}$'s obtained is arranged in a one-dimensional array ($G_i^1, G_i^2, \dots, G_i^k$). In this work we chose $\sigma = 1.0$ and used polynomials with $m = 0, 1, 2, 4, 6$ for 12 $r_0$ values, giving a total of $k = 60$ $G_i^l$'s.

**The BOP potential**. In the BOP model adopted in this work, the energy of an atom $i$ is postulated in the form

$$E_i = \frac{1}{2} \sum_{j \neq i} \left[ e^{A_i - \alpha_i r_{ij}} - S_{ij} b_{ij} e^{B_i - \beta_i r_{ij}} \right] f_c(r_{ij}) + E_i^{(p)}, \quad (6)$$

where $r_{ij}$ is the distance between atoms $i$ and $j$ and the summation is over all atom $j$ other than $i$ within the cutoff radius $r_c$. The bond-order parameter $b_{ij}$ is taken in the form

$$b_{ij} = (1 + z_{ij})^{-1/2}, \quad (7)$$

where

$$z_{ij} = a_i^2 \sum_{k \neq i,j} S_{ik} (\cos\theta_{ijk} + h_i)^2 f_c(r_{ik}) \quad (8)$$

represents the number of chemical bonds (other than $ij$) formed by atom $i$. Larger $z_{ij}$ values (more bonds) lead to a smaller $b_{ij}$ and thus weaker $ij$ bond.

The screening factor $S_{ij}$ reduces the strength of bonds by surrounding atoms. For example, when counting the bonds in Eq. (8), we screen them by $S_{ik}$, so that strongly screened bonds contribute less to $z_{ij}$. The screening factor $S_{ij}$ is given by

$$S_{ij} = \prod_{k \neq i,j} S_{ijk}, \quad (9)$$

where the partial screening factor $S_{ijk}$ represents the contribution of a neighboring atom $k$ (different from $i$ and $j$) to the screening of the bond $ij$. $S_{ijk}$ is given by

$$S_{ijk} = 1 - f_c(r_{ik} + r_{jk} - r_{ij}) e^{-\lambda_i^2 (r_{ik} + r_{jk} - r_{ij})}. \quad (10)$$

It has the same value for all atoms $k$ located on the surface of an imaginary spheroid whose poles coincide with the atoms $i$ and $j$. For all atoms $k$ outside this cutoff spheroid, on which $r_{ik} + r_{jk} - r_{ij} = r_c$, we have $S_{ijk} = 1$ — such atoms are too far away to screen the bond. If an atom $k$ is placed on the line between the atoms $i$ and $j$, we have $r_{ik} + r_{jk} - r_{ij} = 0$ and $S_{ijk}$ is small — the bond $ij$ is strongly screened (almost broken) by the atom $k$. This behavior reasonably reflects the nature of chemical bonding.

Finally, the promotion energy $E_i^{(p)}$ is taken in the form

$$E_i^{(p)} = -\sigma_i \left( \sum_{j \neq i} S_{ij} b_{ij} f_c(r_{ij}) \right)^{1/2}. \quad (11)$$

For a covalent material, $E_i^{(p)}$ accounts for the energy cost of changing the electronic structure of a free atoms before it forms chemical bonds. For example, for group IV elements, this is the cost of the $s^2p^2 \rightarrow sp^3$ hybridization. On the other hand, $E_i^{(p)}$ can be interpreted as the embedding energy

$$F(\bar{\rho}_i) = -\sigma_i (\bar{\rho}_i)^{1/2} \quad (12)$$

appearing in the EAM formalism[1,2]. Here, the host electron density on atom $i$ is given by $\bar{\rho}_i = \sum_{j \neq i} S_{ij} b_{ij} f_c(r_{ij})$. Due to this feature, this BOP model can be applied to both covalent and metallic systems.

The BOP functions depend on eight parameters $A_i$, $B_i$, $\alpha_i$, $\beta_i$, $a_i$, $h_i$, $\sigma_i$, and $\lambda_i$, which constitute the parameter set $(p_1, \dots, p_m)$ with $m = 8$. The cutoff parameters were fixed at $r_c = 6$ Å and $d = 1.5$ Å.

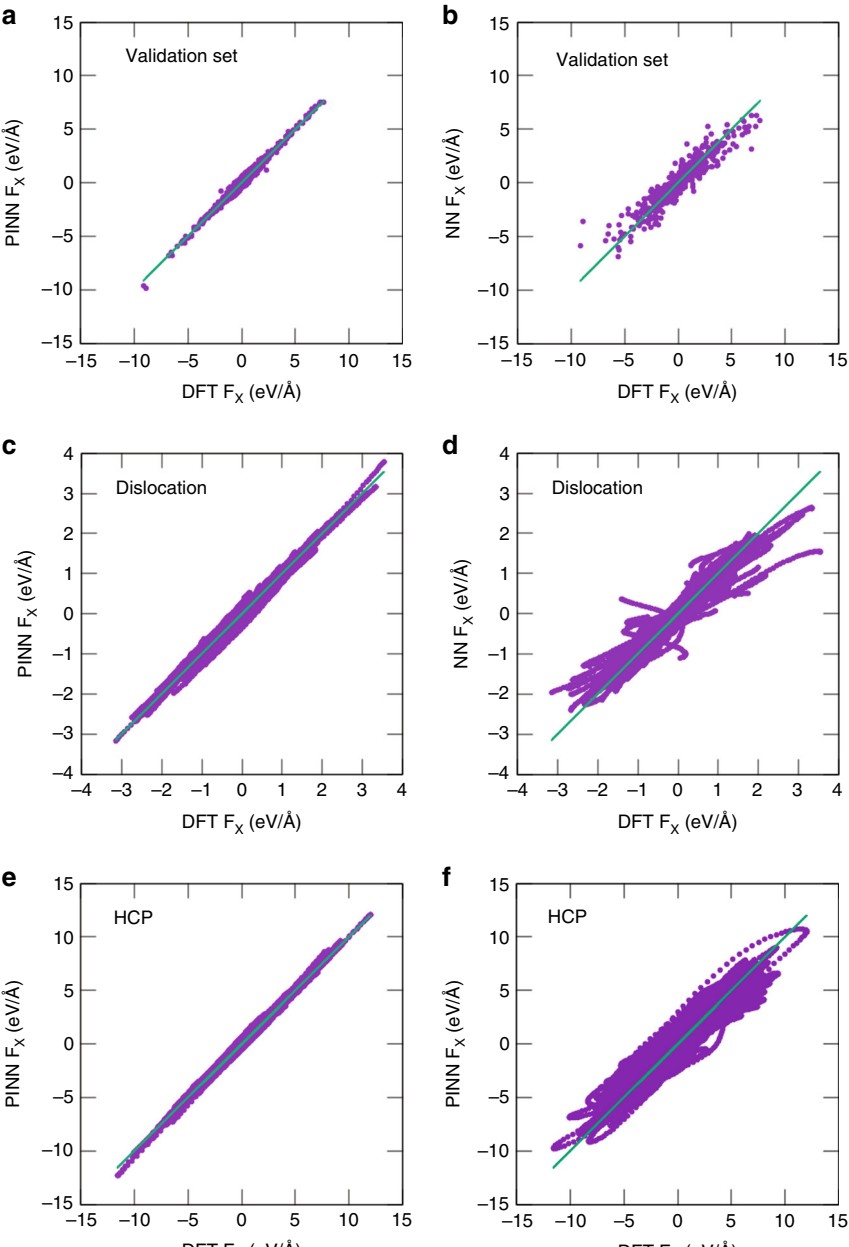

**Fig. 7** Testing of atomic force predictions. The *x*-component of atomic forces for **a**, **b** validation database, **c**, **d** edge dislocation in NVE MD simulations starting at 700 K, and **e**, **f** HCP Al in NVT MD simulations at 300, 600, 1000, 1500, 2000, and 4000 K. The forces predicted by the PINN (**a**, **c**, **e**) and NN (**b**, **d**, **f**) potentials are compared with DFT calculations from refs. [20,21]. The straight lines represent the perfect fit. See Supplementary Figs 11–13 for all components of the forces

**The neural network and training procedures.** The feedforward NN contained two hidden layers and had the $60 \times 15 \times 15 \times 8$ architecture for the PINN potential and $60 \times 16 \times 16 \times 1$ for the NN potential. The number of nodes in the hidden layers was chosen to reach the target accuracy of about 3-4 meV/atom without overfitting.

The training/validation database consisted of DFT total energies for a set of supercells. The DFT calculations were performed using projector-augmented wave (PAW) pseudopotentials as implemented in the electronic structure Vienna Ab initio Simulation Package (VASP)[55,56]. The generalized gradient approximation (GGA) was used in conjunction with the Perdew, Burke, and Ernzerhof (PBE) density functional[57,58]. The plane-wave basis functions up to a kinetic energy cutoff of 520 eV were used, with the *k*-point density chosen to achieve convergence to a few meV per atom level. Further details of the DFT calculations can be found in refs. [20,21]. The energy of a given supercell $s$, $E^s = \sum_i E_i^s$, predicted by the potential was compared with the DFT energy $E_{\mathrm{DFT}}^s$. Note that the original $E_{\mathrm{DFT}}^s$ values were not corrected to remove the energy of a free atom. To facilitate comparison with

literature data, prior to the training all DFT energies were uniformly shifted by 0.38446 eV per atom to match the experimental cohesive energy of Al, 3.36 eV per atom[59]. The NN was trained by adjusting its weights $w_{\epsilon\kappa}$ and biases $b_\kappa$ to minimize the objective function

$$\mathcal{E} = \sum_s \left(E^s - E_{\mathrm{DFT}}^s\right)^2 + \tau\left(\sum_{\epsilon\kappa} |w_{\epsilon\kappa}|^2 + \sum_\kappa |b_\kappa|^2\right) + \gamma\left(\sum_\eta \left|p_\eta - \bar{p}_\eta\right|^2\right). \quad (13)$$

The second term was added to avoid overfitting by controlling the magnitudes of the weights and biases. The parameter $\tau$ controls the degree of regularization. The third term ensures that the variations of the PINN parameters relative to their database-averaged values $\bar{p}_\eta$ remain small. The minimization of $\mathcal{E}$ was implemented by the Davidson–Fletcher–Powell algorithm of unconstrained optimization. The optimization was repeated several times starting from different random states and the solution with the smallest $\mathcal{E}$ was selected as final. The PINN and NN forces were computed by the finite-difference method.

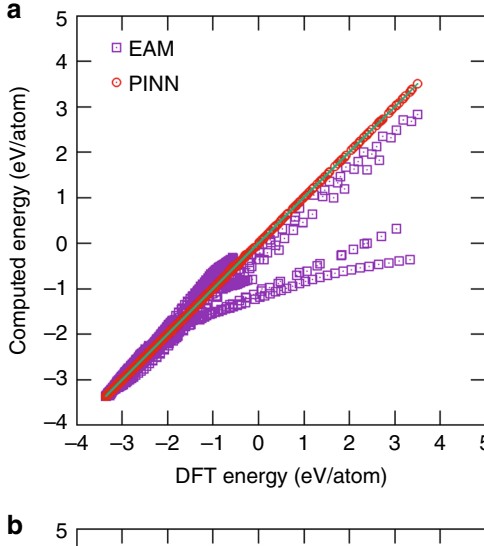

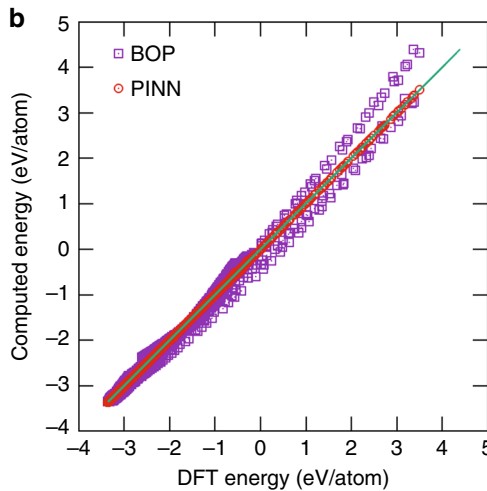

**Fig. 8** Comparison of DFT versus potentials. Energies of atomic configurations in the DFT database used for training and validation are compared with predictions of the **a** EAM Al potential[54] and **b** BOP potential. The BOP parameters were fitted to the DFT database and permanently fixed. The PINN potential predictions are included for comparison. The straight line represents the perfect fit

## Data availability

All data that support the findings of this study are available in the Supplementary Information file or from the corresponding author upon reasonable request.

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

## Acknowledgements

We are grateful to Dr. James Hickman for performing some of the additional Al DFT calculations used in this work. We are also grateful to Dr. Vesselin Yamakov for numerous helpful discussions, the development of a software package for PINN-based simulations, and for benchmarking the computational speed of the method. The authors acknowledge support of the Office of Naval Research under Awards No. N00014-18-1-2612 (G.P.P.P. and Y.M.) and N00014-17-1-2148 (R.B. and R.R.). This work was also supported in part by a grant of computer time from the DoD High Performance Computing Modernization Program at ARL DSRC, ERDC DSRC and Navy DSRC.

## Author contributions

Y.M. developed the PINN theory and initiated this research project. G.P.P.P. wrote the computer software for the NN and PINN potential training, validation and testing under Y.M.'s direction and supervision. He also created the Al NN and PINN potentials reported in this paper and tested their properties. R.B. generated much of the DFT data for Al used in this work under R.R.'s advise and supervision. Y.M. wrote the initial draft of the manuscript. All co-authors were engaged in discussions, contributed ideas at all stages of the work, participated in the manuscript editing, and approved its final version.

## Additional information

**Competing interests:** The authors declare no competing interests.

