## [Peer Review File · Nature Communications]

Reviewers' comments:

Reviewer #1 (Remarks to the Author):

The authors introduce a machine learning model (neural network based) for parameters in bond-order potentials (BOP). They claim unprecedented transferability for the resulting BOP. The claim is substantiated by numerical results for Al.

To the best of my knowledge, the particular combination of machine learning with BOP is novel. The approach presented is sound, and in all likelihood correct. Furthermore, the numerical demonstration and results presented are compelling. The paper is well written, and figures are publication quality. The comparison of PINN to NN is particularly striking and extremely newsworthy to the entire community.

My only criticism is that the authors do not provide proper reference and credit to previous work which already explored some aspects of the underlying idea in this paper. More specifically, the idea to combine physics based potentials with regression models is not new. It was firmly established in the context of molecular dynamics by Alessandro de Vita and co-workers, and the authors should provide additional references to his contributions. In the context of chemical space (the crucial space in which transferability is typically assessed), it was already studied by

*Ramakrishnan et al, JCTC, 2015 (Delta machine learning)

*Dral et al, JCTC, 2015 (ML models of OM2 parameters)

*Bereau et al, JCTC, 2015 (ML models of atomic multipoles)

*Bereau et al, JCP, 2018 (ML models of non-covalent potentials)

*Kranz et al, JCTC 2018 (ML models of parameters in tight binding DFT)

As such, while I think that this paper constitutes a major contribution, these references should be included before I can recommend it for publication in Nature Communications.

Reviewer #2 (Remarks to the Author):

The authors propose a neural network architecture that incorporates physically-informed terms in its output layer, specifically a bond order potential that models attractive and repulsive forces between atoms explicitly, in dependence of their local environments.

Their self-imposed objective is twofold: to address the lack of transferability of ML models, and to improve the behavior of ML methods in the extrapolation regime of the data. Both are pressing issues in the field. The proposition in the ms is that a neural network architecture with physical constraints will generate physically meaningful answers in those challenging edge cases.

They demonstrate the capability of their architecture by reconstructing a potential for Aluminum, which is trained on a database of different crystal structures of varying size and volume and corresponding DFT energy labels. The quality of that potential is evaluated by predicting the energies along an existing MD trajectory of previously unseen structures. Its performance is then compared to a baseline method, which is the same network architecture, but without the subsequent physics layer.

In principle, the manuscript is well written, with a clear structure and an easy to follow flow of arguments. However the argumentation appears flawed. Specifically:

The ms says, "The proposed PINN model is based on the following considerations. A traditional, physics-based potential can always be trained to reproduce the energy of any given atomic configuration with any desired accuracy. Of course, this potential will not work well for other configurations."

The implication is that merely introducing a physically motivated output layer solves the model selection problem, which then would only leave the transferability problem to be solved. The latter is tackled by arguing that their NN feeds its output into the bond order potential and thus acts as a

dynamic parametrization for it.

There are several issues with this argumentation:

The first statement is only true if the potential is unbiased in the sense that it models a function class that in fact contains the perfect solution. A harmonic oscillator, for example, is "physics-based" but unable to accurately model complex atomic interactions. On the other hand, the universal approximation theorem tells us that any neural network with one or more hidden layer is an universal approximator.

In fact, most recent ML models for potential energy surfaces use universal approximators. What sets them apart is how quickly they reach a certain accuracy. Two general approximators might behave vastly different in terms of efficiency: one might need several order of magnitude less training examples to achieve the same performance. The efficiency of an approximator determines its usefulness, not its theoretical ability to model the problem.

Secondly, the statement of a NN that dynamically parametrizes a proven potential is compelling, however it is never demonstrated that this actually works effectively. In fact, the published experimental results show only minor improvements over their baseline method without the physical potential. Furthermore, the authors never compare to state-of-the-art approaches with modern architectures that go beyond the basic fully-connected two layer layout of their baseline (i.e. convolutional networks, message passing, resnet that are part of a broad literature also applied to MD). In the same spirit it would be illuminating to get an idea of the performance of their method on more popular datasets.

Finally, the stated purpose of their development is to accelerate MD simulations, which they never perform with their model. The authors only predict energies and numerical forces for existing trajectories. This is unfortunate, because I would expect their model to be very robust in that particular application due to the physical constraints and I believe this could be one of the main selling points.

The authors should aim for a broader embedding into the literature as several relevant publications that address similar problems using related approaches are missed. :

- Nat. Comms., 9(1), 2018, 3887
- JCP 148, 241709 (2018)
- JCP 148, 241722 (2018)
- JCTC 2017 13(11), 5255-5264
- arXiv: 1704.01212 (2017)

In the light of the above shortcomings, I think the paper is not suitable for publication.

Reviewer #3 (Remarks to the Author):

The authors propose a combined model (PINN) of a neural network (NN) which is used to determine the parameters of an analytical bond-order potential (BOP). The rationale is that the BOP is "physically-informed" about the functional form of the interatomic potential, while the NN yields the flexibility to achieve a high accuracy. This is supposed to improve the capability to extrapolate beyond the training domain. In a similar manner, machine learning has previously been applied to parametrize physics-based potential (Bereau et al 2018 JCP). While the approach sounds appealing, I have several fundamental concerns with the manuscript at the current stage.

The authors state that a "purely mathematical extrapolation scheme" is unlikely to yield physically meaningful results. However, the combined model also has to degrade in accuracy in the extrapolation domain, since the NN parametrization of the BOP is no longer informed by data, but only on the functional form of the original, physical potential. The manuscript shows no proof that the performance of PINN exceeds that of the original BOP in the extrapolation domain. In fact, the

performance might even be worse in this case, due to the added flexibility from the NN parametrization. This possible degradation of accuracy should be evaluated in detail.

For instance, in Fig. 4, where extrapolation w.r.t. the cell volume is shown, the development of the PINN and NN potentials does not look like a smooth continuation of the DFT potential. When entering the extrapolation domain, the predicted PINN potential is rather rapidly constrained to approach zero. Therefore, it is difficult to judge which of the ML potentials performs better in this area. For a proper evaluation, DFT calculations should be performed here. Constraining PINN to approach zero at the BOP cutoff does not necessarily make it a better extrapolation, but might also simply hint at a cutoff that was chosen too small. At the very least, the extrapolation performance should be compared to the original potential.

The improvements of PINN over NN seem to be hit and miss. While the inaccuracies of the NN in the extrapolation domain might sometimes be larger, e.g. regarding force predictions in Fig. 7, the PINN still exhibits large errors of 0.1eV/Å such that the improvement might not be enough to achieve DFT-level results in a simulation. Considering that the dataset only contains Al and >3000 reference calculations were used for training, a much more effective strategy would be to reduce the required amount of training data with established methods (e.g. force training; Pukrittayakamee et al 2009 JCP, Chmiela et al 2017 Sci Adv, Glielmo et al 2017 PRB) and spend the saved computation time to add reference calculations in the current "extrapolation domain", e.g. using query-by-committee or on-the-fly learning (Gastegger et al 2017 (Chem Sci), Li et al 2015 (PRL)). The proposed PINN approach might even be obstructing the detection of the extrapolation domain, since models of a NN ensemble would be constrained to a (possibly inaccurate) functional form. Therefore, I am not convinced of the applicability of the proposed approach at the current stage for the purpose of achieving DFT-accurate interatomic potentials.

Further issues:

- The authors give training and test errors in Table I, however, the NN architecture and hyper-parameters were optimized on the test set. It is common practice to use a separate validation set for this purpose and test on unseen data. Beyond that, the test error of PINN is lower than the training error for PINN. This is unusual and might be due to a "lucky split" of the dataset. For a thorough evaluation, the experiments should be repeated on different random splits and standard errors should be given.
- In the same table, the NN has a higher training error than PINN while having less parameters. Why did the authors not compare to a model with the same number of parameters? Would increasing the number of parameters of NN, increase the test error?
- The manuscript only demonstrates the method on Al structures. For a proper demonstration of the method, this should also be demonstrated on systems containing multiple elements. Since the NN is a modified Behler-Parinello network, this should be straight-forward.
- The manuscript is missing several relevant references, e.g. to modern end-to-end architectures such as DTNN (Schütt et al, Nat Comm 2017), MPNN (Gilmer et al ICML 2017), SchNet (Schütt et al, JCP 2018) and HIP-NN (Lubbers et al, JCP 2018).
- It is not quite clear to me at which level of theory the reference data was calculated. It seems to be a mixture of XC functionals? For an evaluation of the proposed method it might be better restrict the model to one level of theory.

Concluding, I am not fully convinced that the proposed PINN is "physically-informed" beyond being constrained to approach zero at the cutoff. The authors do neither demonstrate that their approach is general regarding the composition of the training data nor reach sufficient accuracy in the extrapolation domain. Therefore, I cannot recommend this manuscript for publication at this stage.

way.

Response to Reviewers' comments

Re: Manuscript NCOMMS-18-27853-T

“Physically-informed artificial neural networks for atomistic modeling of materials”

by G. P. Purja Pun, R. Batra, R. Ramprasad and Y. Mishin

November 12, 2018

We are grateful to all three Referees for providing insightful comments on our paper. This document summarizes our responses to the Referees and the changes made in manuscript. Such changes include additional technical information, rephrasing of several paragraphs for clarity, replacement of Fig.4 by two new figures, updated versions of Figs. 2 (c,d) and S1, and 7 new literature references [26, 27, 31, 38, 45, 46, 47] (highlighted in yellow). A few typing errors have been corrected. We have also swapped the terms “testing” and “validation” to comply with the more conventional usage of these terms with the machine learning community.

Before we address the specific comments and questions from the Reviewers, we would like to make some general comments.

All three Reviewers recommended that we include additional citations of previous literature. We did include several of these citations. However, looking at the recommended literature and some of the comments/questions, we realize that the Reviewers likely work in the field of computational chemistry. As such, they tend to evaluate our paper from the standpoint of that field. The recommended papers mostly deal with molecular (typically, organic) matter and are focused on predicting properties and trends across diverse molecular structures and wide ranges of chemical compositions, often encompassing thousands (in one of the papers, hundreds of thousands) of different molecules. The Reviewers emphasize the importance of the compositional space and expect that a transferable potential should be a molecular field tested on a large compositional domain.

We would like to emphasize that our paper lies in the field of computational materials science rather than computational chemistry. In this field, we are focused on the configurational (rather than chemical) space of atomic systems. We deal primarily with crystal structures and occasionally liquids, focusing the attention on lattice defects, such as vacancies, interstitials, grain boundaries, stacking faults, dislocations, surfaces, cracks, etc, and their interactions. The main goal is to describe mechanical and thermal properties of the material. Most of the interatomic potentials in this field are for single-component systems. Binary potentials are expected to correctly predict thermodynamic properties (e.g., the phase diagram), and are used to model solute segregation, solute drag on grain boundaries and dislocations, and their impact on deformation behavior and fracture. Accurate ternary potentials remain a dream. A good interatomic potential is expected to be general-purpose type: once created, it is used for almost any type of MD simulations, often well outside the training domain. Successful general-purpose potentials are used by the community for 10-20 years, sometimes longer. There are obviously significant cultural differences between the two fields, and thus a different understanding of what constitutes a breakthrough in the field.

Most of the traditional interatomic potentials in our field are based on physical intuition for a particular material or class of materials. The best potentials do in fact deliver physically meaningful results for a wide range of properties and are widely used by the community. Their weakness is in the limited accuracy. In extrapolation domains, deviations from first-principles calculations can be significant. On the other hand, there is a growing number of papers

attempting to construct machine-learning potentials. After a period of excitement, the community seems to realize that such potentials can only be helpful for particular types of applications but are unlikely to become general-purpose potentials as expected in this field. While reaching the DFT level of accuracy during the training, such potentials can fail miserably when used outside the training domain.

In this paper we propose a new approach that takes the best from each side. The PINN potentials are as accurate as the mathematical machine-learning potentials within the training domain. At the same time, like the traditional physics-based potentials, they continue to deliver rather accurate, or worst case at least physically meaningful results, in extrapolation domains of the atomic configuration space. Technical details of the method are in the paper. The PINN AI potential presented in the paper is only included to demonstrate the principle. We are currently working on similar potentials for other elements and on generalizing the PINN model to binary and later multi-component systems. Software development for the new method is also in progress.

We believe that the development of the PINN potentials constitutes a very significant advance in this field. We foresee that this work will create a wave that will soon propagate across the field and transform it by raising the reliability of the interatomic potentials to a new level. The goal of this paper is to make the first announcement of the method. It will be followed by further publications related to further improvements and specific materials.

We ask the Reviewers to consider the differences between the two fields mentioned above when evaluating the revised version of the manuscript.

Reviewer #1:

Reviewer:

The authors introduce a machine learning model (neural network based) for parameters in bond-order potentials (BOP). They claim unprecedented transferability for the resulting BOP. The claim is substantiated by numerical results for AI.

To the best of my knowledge, the particular combination of machine learning with BOP is novel. The approach presented is sound, and in all likelihood correct. Furthermore, the numerical demonstration and results presented are compelling. The paper is well written, and figures are publication quality. The comparison of PINN to NN is particularly striking and extremely newsworthy to the entire community.

My only criticism is that the authors do not provide proper reference and credit to previous work which already explored some aspects of the underlying idea in this paper. More specifically, the idea to combine physics based potentials with regression models is not new. It was firmly established in the context of molecular dynamics by Alessandro de Vita and co-workers, and the authors should provide additional references to his contributions. In the context of chemical space (the crucial space in which transferability is typically assessed), it was already studied by

**Ramakrishnan et al, JCTC, 2015 (Delta machine learning)*

**Dral et al, JCTC, 2015 (ML models of OM2 parameters)*

**Bereau et al, JCTC, 2015 (ML models of atomic multipoles)*

**Bereau et al, JCP, 2018 (ML models of non-covalent potentials)*

**Kranz et al, JCTC 2018 (ML models of parameters in tight binding DFT)*

Response:

We are grateful to the referee for the appreciation of the novelty and soundness of our approach and its noteworthiness to the entire community.

We are aware of the recent work by late Alessandro de Vita and co-workers, in particular their very interesting papers where some of the interaction terms (e.g., the multipole expansion parameters) are fitted using regression models. We have added references to such papers to put our work in perspective. However, to the best of our knowledge, combining a sophisticated interatomic potential model (BOP in our case) with ML-based predictions of its parameters and obtaining an accurate general-purpose force field is a new step that has not been reported in the literature before.

We should also note that broad exploration of the chemical space mentioned by the Reviewer is not our goal in this work or in the near future. We are mostly focused on exploration of the configurational space of single-component systems, planning to extend our approach to binary systems soon and hopefully ternary systems in the future. While we appreciate that the chemical space is “the crucial space in which transferability is typically assessed” in the computational chemistry community, our work lies in a somewhat different field where transferability is understood as the ability to predict the energy and forces for atomic configurations outside the configurational domain represented by the training dataset. The literature mentioned by the Reviewer mostly explores the chemical space. Some aspects are still relevant to our work, and we did include some of the recommended citations, but not all.

Reviewer:

As such, while I think that this paper constitutes a major contribution, these references should be included before I can recommend it for publication in Nature Communications.

Reviewer #2:

Reviewer:

The authors propose a neural network architecture that incorporates physically-informed terms in its output layer, specifically a bond order potential that models attractive and repulsive forces between atoms explicitly, in dependence of their local environments.

Their self-imposed objective is twofold: to address the lack of transferability of ML models, and to improve the behavior of ML methods in the extrapolation regime of the data. Both are pressing issues in the field. The proposition in the ms is that a neural network architecture with physical constraints will generate physically meaningful answers in those challenging edge cases.

They demonstrate the capability of their architecture by reconstructing a potential for Aluminum, which is trained on a database of different crystal structures of varying size and volume and corresponding DFT energy labels. The quality of that potential is evaluated by predicting the energies along an existing MD trajectory of previously unseen structures. It's performance is then compared to a baseline method, which is the same network architecture, but without the subsequent physics layer.

In principle, the manuscript is well written, with a clear structure and an easy to follow flow of arguments. However the argumentation appears flawed. Specifically:

The ms says, “The proposed PINN model is based on the following considerations. A traditional, physics-based potential can always be trained to reproduce the energy of any given atomic configuration with any desired accuracy. Of course, this potential will not work well for other configurations.”

The implication is that merely introducing a physically motivated output layer solves the model selection problem, which then would only leave the transferability problem to be solved. The latter is tackled by arguing that their NN feeds its output into the bond order potential and thus acts as a dynamic parametrization for it.

There are several issues with this argumentation:

The first statement is only true if the potential is unbiased in the sense that it models a function class that in fact contains the perfect solution. A harmonic oscillator, for example, is “physics-based” but unable to accurately model complex atomic interactions. On the other hand, the universal approximation theorem tells us that any neural network with one or more hidden layer is an universal approximator.

Response:

This is not what we are claiming in the first statement. We are not claiming that the BOP can perfectly, or very accurately, approximate the entire training database of DFT energies. All we are saying in this sentence is that the model can accurately reproduce the DFT energy of one given structure, which is just one number. This can be trivially achieved by adjusting the repulsion/attraction coefficients A and B in equation (6) of the Supplementary file while assigning arbitrary values to the remaining coefficients. So this statement is perfectly correct. We are further stating that this set of parameters will not work well for other atomic configurations, which is again obvious. This reasoning does not involve a model selection problem or a search for a perfect (or just accurately enough) solution.

Reviewer:

In fact, most recent ML models for potential energy surfaces use universal approximators. What sets them apart is how quickly they reach a certain accuracy. Two general approximators might behave vastly different in terms of efficiency: one might need several order of magnitude less training examples to achieve the same performance. The efficiency of an approximator determines its usefulness, not its theoretical ability to model the problem.

Response:

The universal approximation theorem guarantees accurate fit to a given training set but not necessarily reasonable transferability to unknown configurations in distant domains of the configuration space. In this sense, it could be called a “universal interpolation” theorem without any claim of extrapolation ability. This is exactly the problem that we are addressing in this paper. The proposed combination of NN and BOP still guarantees infinitely accurate approximation of any given training dataset. The sets of BOP parameters that each perfectly match the individual DFT energies (see our comment above) constitutes the new training set for the NN part of the architecture. This is clearly stated in the paper. Thus, the PINN potential does work as a universal approximator. Its speed of convergence during the training is not our primary concern at this point. Our main goal is to achieve a physically meaningful extrapolation, not just accurate interpolation.

Reviewer:

Secondly, the statement of a NN that dynamically parametrizes a proven potential is compelling, however it is never demonstrated that this actually works effectively. In fact, the published experimental results show only minor improvements over their baseline method without the physical potential.

Response:

We are not sure what the Reviewer means by “experimental results”. Experiments are only discussed in the paper in the context of thermal expansion, for which the PINN potential shows excellent agreement and performs much better than the straight NN potential (Fig. 5). Table II compares the NN and PINN predictions with existing DFT calculations for selected properties, and the PINN potential again shows a substantial improvement. But the most convincing demonstration of the improved transferability of the PINN model comes from the independent tests performed on atomic configurations that were not in the training or validation datasets and represent significantly different atomic environments. Numerous examples are presented in Figures 4, 6 and 7 of the main text and S6-S13 of the Supplementary Information file.

Reviewer:

Furthermore, the authors never compare to state-of-the-art approaches with modern architectures that go beyond the basic fully-connected two layer layout of their baseline (i.e. convolutional networks, message passing, resnet that are part of a broad literature also applied to MD). In the same spirit it would be illuminating to get an idea of the performance of their method on more popular datasets.

Response:

We are not sure what the Review means by “more popular datasets”. Regarding the modern architectures, we may try convolutional networks in the future. At the present stage, it is not our goal to maximize the accuracy or computational performance of the baseline model (i.e., the NN itself). We know that we can easily achieve the training accuracy on the 1 meV/atom level using a standard feed-forward NN. The whole point is that we are moving away from the purely mathematical NN models of any architecture since they are not transferable. The proposed PINN model does contain a NN, but it only plays a supporting role for the BOP potential. At this stage of the work, our goal is to demonstrate the principle. Using a standard feedforward NN perfectly serves this purpose.

Reviewer:

Finally, the stated purpose of their development is to accelerate MD simulations, which they never perform with their model. The authors only predict energies and numerical forces for existing trajectories. This is unfortunate, because I would expect their model to be very robust in that particular application due to the physical constraints and I believe this could be one of the main selling points.

Response:

Our goal was not to “accelerate” MD simulations in comparison with the mathematical ML (e.g., NN) potentials. In fact, the addition of the BOP step can make the energy and force calculations slower (but, of course, still orders of magnitude faster in comparison with straight DFT calculations). Our main goal is to make MD simulations more reliable and applicable to more diverse simulation conditions while keeping the same DFT-level of accuracy as much as possible. Our progress toward this goal has been demonstrated by the tests for the energies and atomic forces shown in the paper.

We agree with the Reviewer that the PINN model is expected to be very robust due to the physical constraints. This has indeed been confirmed by the tests reported in the paper. To make PINN-based MD simulations accessible to the broader materials modeling community, we are planning to incorporate the PINN model in interatomic potential repositories such as the OpenKim Project (<https://openkim.org>) and the NIST Interatomic Potentials Repository (<https://www.ctcms.nist.gov/potentials/>). In the future, we will also implement the PINN model in popular MD software such as LAMMPS (<https://lammps.sandia.gov>) and will interface it with the Python-based Atomic Simulation Environment (<https://wiki.fysik.dtu.dk/ase/index.html>).

Reviewer:

The authors should aim for a broader embedding into the literature as several relevant publications that address similar problems using related approaches are missed. :

- *Nat. Comms.*, 9(1), 2018, 3887
- *JCP* 148, 241709 (2018)
- *JCP* 148, 241722 (2018)
- *JCTC* 2017 13(11), 5255-5264
- *arXiv:1704.01212* (2017)

In the light of the above shortcomings, I think the paper is not suitable for publication.

Response:

As in the case of Reviewer #1, the suggested papers mostly deal with ML exploration of the compositional space in the context of computational chemistry. This is not exactly the field for which the proposed PINN potentials are intended. But we have included two of the suggested papers as general reference.

Reviewer #3:

Reviewer:

The authors propose a combined model (PINN) of a neural network (NN) which is used to determine the parameters of an analytical bond-order potential (BOP). The rationale is that the BOP is "physically-informed" about the functional form of the interatomic potential, while the NN yields the flexibility to achieve a high accuracy. This is supposed to improve the capability to extrapolate beyond the training domain. In a similar manner, machine learning has previously been applied to parametrize physics-based potential (Bureau et al 2018 JCP). While the approach sounds appealing, I have several fundamental concerns with the manuscript at the current stage.

Response:

We have included two citations of Bureau et al (2015, 2018). But again, those papers deal with ML parametrization of a simple intermolecular interaction model for multi-component systems across a broad chemical space, focusing on organic and biological systems, such as amino acids and DNA. We propose a more rigorous (BOP) interatomic interaction model for single-component (in the future binary) condensed matter with parameters adjusted locally by a neural network. The focus of our work is not to just parameterize but to ensure an improved transferability of the model to extrapolation domains. This approach is significantly different.

Reviewer:

The authors state that a "purely mathematical extrapolation scheme" is unlikely to yield physically meaningful results. However, the combined model also has to degrade in accuracy in the extrapolation domain, since the NN parametrization of the BOP is no longer informed by data, but only on the functional form of the original, physical potential.

The manuscript shows no proof that the performance of PINN exceeds that of the original BOP in the extrapolation domain. In fact, the performance might even be worse in this case, due to the added flexibility from the NN parametrization. This possible degradation of accuracy should be evaluated in detail.

For instance, in Fig. 4, where extrapolation w.r.t. the cell volume is shown, the development of the PINN and NN potentials does not look like a smooth continuation of the DFT potential. When entering the extrapolation domain, the predicted PINN potential is rather rapidly constrained to approach zero. Therefore, it is difficult to judge which of the ML potentials performs better in this area. For a proper evaluation, DFT calculations should be performed here. Constraining PINN to approach zero at the BOP cutoff does not necessarily make it a better extrapolation, but might also simply hint at a cutoff that was chosen too small. At the very least, the extrapolation performance should be compared to the original potential.

Response:

We are not sure what the Reviewer means by the "original potential". The DFT calculations? But the Reviewer does make a very good point. The accuracy of the PINN model may indeed degrade as we deviate further away from the training configurations. No model can possibly keep the same DFT-level accuracy across the entire configuration space. From our experience so far,

the loss of accuracy is slow, and most importantly, the results still remain physically meaningful. Even if the BOP parameters deviate from optimal values, the potential still continues to predict repulsion at short atomic separations, bonding at larger separations and eventually decrease in the bonding energy, the weakening of bonds with increasing number of neighbors, bond screening by the surrounding atoms, and other physical effects built into the BOP model. Qualitatively, the behavior will still be physical, although the numerical accuracy can diminish. (By contrast, a purely mathematical model can produce total nonsense in the extrapolation domain).

Specifically for the Al PINN potential, the numerous tests reported in the paper (e.g., Figures 4, 6, 7, S6-S13) show that the PINN potential does not always predict perfectly accurate results in the extrapolations domain, but the predictions remain physically meaningful and are much more accurate than those of the mathematical NN potential.

Regarding specifically Fig.4, we agree that we have not chosen the most convincing demonstration of the possible failures of the NN potential. We have replaced this figure by two other plots that we believe provide a better demonstration. We compress the crystal structures to atomic volumes smaller than any volume used for training or validation. The PINN potential correctly predicts stiff repulsion, with energies accurately reproducing the DFT points that were computed after this work was complete. By contrast, the NN potential immediately develops uncontrollable wiggles and unphysical behavior.

Reviewer:

The improvements of PINN over NN seem to be hit and miss. While the inaccuracies of the NN in the extrapolation domain might sometimes be larger, e.g. regarding force predictions in Fig. 7, the PINN still exhibits large errors of 0.1eV/Å such that the improvement might not be enough to achieve DFT-level results in a simulation. Considering that the dataset only contains Al and >3000 reference calculations were used for training, a much more effective strategy would be to reduce the required amount of training data with established methods (e.g. force training; Pukrittayakamee et al 2009 JCP, Chmiela et al 2017 Sci Adv, Glielmo et al 2017 PRB) and spend the saved computation time to add reference calculations in the current "extrapolation domain", e.g. using query-by-committee or on-the-fly learning (Gastegger et al 2017 (Chem Sci), Li et al 2015 (PRL)). The proposed PINN approach might even be obstructing the detection of the extrapolation domain, since models of a NN ensemble would be constrained to a (possibly inaccurate) functional form. Therefore, I am not convinced of the applicability of the proposed approach at the current stage for the purpose of achieving DFT-accurate interatomic potentials.

Response:

There are two statements here. One is related to the forces. Indeed, we could have improved the accuracy of the force predictions by including them (in addition to energies) in the objective function during the training (the well-known force-matching approach). This is in our plans for the future PINN potentials. In this paper, we did not do this on purpose, since we wanted to check how the PINN potential performs in predicting forces without fitting to them in comparison with the NN potential. Figure 7 demonstrates convincingly that the predictive capability of PINN is much better.

In the second point, the Reviewer suggests that we could have included the "extrapolation domain" configurations during the training. Sure we could, but then it would not be an extrapolation domain anymore. If we encountered another extrapolation domain during the subsequent MD simulations, a mathematical model would again produce unpredictable nonsense. One can try to automatically detect when we enter an extrapolation domain during the simulation, but at that point we would have to abort the simulation anyway. One can also try to learn (train) in such domains on the fly, but this cannot continue forever, and in any case this is not the direction we are moving in. Our approach is to ensure that, whenever we enter an

extrapolation domain, we continue to obtain meaningful results, even though the accuracy may eventually deteriorate if we go too far. Even in the latter case, it is still better to sacrifice some of the DFT-level accuracy but keep the simulation running than to start generating nonsensical results as soon as we enter an uncharted territory. In the future, we may add a smooth filter between the NN and the BOP suppressing large deviations of the potential parameters from the average values found during the training.

Reviewer:

Further issues:

- The authors give training and test errors in Table I, however, the NN architecture and hyper-parameters were optimized on the test set. It is common practice to use a separate validation set for this purpose and test on unseen data.

Response:

This comment was prompted by us using the terms “testing” and “validation” in the opposite order (as it is done in some of the papers on interatomic potentials). In the resubmission, we have swapped the two terms to comply with the more established terminology. Our model was optimized on the training dataset. A relatively small validation set was used to monitor the validation error and make sure it does not diverge from the training error, which would signal overfitting. Once the training was complete, the model predictions were compared with independent DFT calculations that were not seen during the training or validation - exactly as described by the Reviewer. The comparison is shown in Figures 4, 6 and S6-S10.

Reviewer:

Beyond that, the test error of PINN is lower than the training error for PINN. This is unusual and might be due to a “lucky split” of the dataset. For a thorough evaluation, the experiments should be repeated on different random splits and standard errors should be given.

Response:

We agree that the lower validation error in this Table is a result of a “lucky split”. Indeed, the validation dataset that we used during the training (to control overfitting) was relatively small (see table S2). At this point of the software development, the training of the PINN model remains computationally expensive. We are not able to repeat the training process multiple times with different splits. Instead, we estimated the final validation error by averaging over a much larger set of DFT energies, drawn from the existing database [20,21], that were not used for training or validation during the training. The new histograms of such error are shown in Figures 2(c,d) and S1. The validation errors appearing in Table I have been undated. As expected, the validation errors based on the better statistics are larger than the respective training errors.

Reviewer:

- In the same table, the NN has a higher training error than PINN while having less parameters. Why did the authors not compare to a model with the same number of parameters? Would increasing the number of parameters of NN, increase the test error?

Response:

The NN and PINN are different types of model whose “fair” comparison is hard to define. The neural networks have different architectures: a single output node in the mathematical NN model and 8 nodes (number of BOP parameters) in the PINN model. As a crude metric for comparison, we chose the total number of fit parameters: 1265 and 1283, respectively (1.4% difference). We then trained each model to approximately the same error. As shown in Table I, the NN model does happen to have a slightly higher training error (4.24 meV versus 4.03 meV, 5% difference). We do not believe these small differences have any significance. We could have made a few more NN training iterations and reached exactly the same error of 4.03 meV as for the PINN

model. This would not have changed the performance of the NN model or the conclusions of the paper. Comparison of the two different models simply on the basis of the number of parameters is somewhat arbitrary anyway. This was only done here to give the reader a general idea about the degree of superior performance of the PINN model.

Reviewer:

- The manuscript only demonstrates the method on AI structures. For a proper demonstration of the method, this should also be demonstrated on systems containing multiple elements. Since the NN is a modified Behler-Parinello network, this should be straight-forward.

Response:

The PINN model is different from the Behler-Parrinello method in a number of ways, from different descriptors to the addition of the BOP part. Generalization to multicomponent systems requires very significant changes, including re-derivation and coding of the new forces for running MD simulations. Most importantly, we are implementing the so-called “inheritance” feature for multicomponent potentials, which is not part of the Behler-Parrinello model, or any other model that we know of. Namely, the traditional EAM, MEAM, ADP and similar potentials in our field are usually obtained by first creating good-quality single-component potentials, say for components A and B, and then “crossing” them by fitting the A-B interaction parameters only using a database of binary supercells. This allows us to avoid dealing with multiple potentials for the same element, depending on whether it is part of an A-B binary, A-C binary, or a separately trained elemental potential A. We are working to formulate a multicomponent PINN model by combining (“inheriting”) existing elemental PINNs. Given that this method will also need to be implemented in massive parallel software, this is a major undertaking that takes time. In the meantime, the purpose of this paper is to present (announce) the general idea of the PINN method and demonstrate the principle by the AI PINN potential. Work is well underway to construct PINN potentials for several other elements before “crossing” them to binary systems.

Reviewer:

- The manuscript is missing several relevant references, e.g. to modern end-to-end architectures such as DTNN (Schütt et al, Nat Comm 2017), MPNN (Gilmer et al ICML 2017), SchNet (Schütt et al, JCP 2018) and HIP-NN (Lubbers et al, JCP 2018).

Response:

We have included some of these references. But as in our responses to Reviewers # 1 and 2, we note that most of these papers discuss exploration of the compositional space in molecular systems. This paper has a different goal: we are more focused on configurational degrees of freedom in atomic systems, and our main concern is physics-based transferability to unknown configurations rather than accurate and efficient training on a given (or growing) reference database.

-Reviewer:

It is not quite clear to me at which level of theory the reference data was calculated. It seems to be a mixture of XC functionals? For an evaluation of the proposed method it might be better restrict the model to one level of theory.

Response:

The reference energies were obtained by DFT calculations using the PBE exchange-correlation functional implemented in VASP. We have added further details in the Methods section.

Reviewer:

Concluding, I am not fully convinced that the proposed PINN is "physically-informed" beyond being constrained to approach zero at the cutoff. The authors do neither demonstrate that their approach is general regarding the composition of the training data nor reach sufficient accuracy in

the extrapolation domain. Therefore, I cannot recommend this manuscript for publication at this stage.

Response:

We hope that the comments presented above and the changes made to the manuscript will clarify all issues and convince the Reviewers that the paper deserves to be published in the Nature Communications.

Sincerely,
G. P. Purja Pun, R. Batra, R. Ramprasad and Y. Mishin.

Reviewers' comments:

Reviewer #2 (Remarks to the Author):

I appreciate the revision, however, I find it disappointing. I feel that the authors have mainly try to argue my comments away than acting upon them properly. This is a pity as I am still not convinced. Comparison to state of the art architectures would help the paper. Clearly I would like to see the system used in an MD simulation, demonstrating the success of their approach convincingly and not alluding to its potential. Also the argument of the on the fly computations was not properly addressed. Concluding, I still cannot recommend publication of the paper in its current form.

Reviewer #3 (Remarks to the Author):

The manuscript has been improved, some of the issues have been resolved and some relevant work was cited. The authors remark that much of the related work suggested work is not relevant due to its application in computationally chemistry. Here, I strongly disagree in that the methods from the mentioned literature are very similar from a machine learning perspective, independent of their respective application. I also disagree with the statement of the authors that on-the-fly learning would "continue forever": e.g. using ensembles as uncertainty estimates, no further calculation will be required, once the simulation has sufficiently explored the relevant domain.

My general criticism remains that PINN might obtain qualitatively "meaningful" results in the extrapolation domain, but with a seriously decreased accuracy. Since I am not too familiar with the accuracy requirements in materials science, I have to trust that this is good enough for the desired application. Beyond that, the authors have not addressed some of the raised issues sufficiently:

- Evaluation of degraded accuracy in extrapolation domain: Does PINN extrapolate better/worse than BOP? Adding BOP to some of the key figures and tables, e.g. Fig. 4 and Table I, would help to judge this.
- Repetitions of training on different splits: the authors state in their response that they "are not able to repeat the training process multiple times with different splits". This raises serious concerns regarding the computational cost of PINN. A neural network of this size using around 3,000 training instances should usually be trained within a couple of hours. The authors should give the computation times for training and prediction of PINN.

Reviewer #4 (Remarks to the Author):

In this work, the authors have developed a physically-informed neural network (PINN) method that uses neural networks to parameterize classical force fields. The key potential benefit of this approach is that such a combined method could have the advantage of both machine learning force fields (accuracy) and classical force fields (transferability). Transferability is an important problem for current machine learning (ML) based force fields. PINN as a general approach could be impactful if it does indeed possess the advantages claimed by the authors.

However, in reading through the paper and the other referee reports, I agree with Reviewer #2 that the manuscript lacks some key comparisons that are needed in order to make the claim of combined accuracy and transferability.

First, the baseline is weak. The authors compare PINN with a fully-connected NN with the same architecture without the BOP. The improved accuracy and transferability are not surprising because fully-connected NN are known to be prone to overfitting. The authors should compare

PINN with more recent ML forces field packages (like PROPhet) and report the improvements in accuracy and transferability based on such comparisons.

Second, the evidence of improved transferability is lacking. Transferability means that the model can extrapolate to a part of phase space not covered by the training data. The only evidence of such transferability in the manuscript is in Fig. 4, where the energy is correctly predicted outside the region of training data. However, this is in fact guaranteed by the form of BOP. This means that even if the output of the neural network is a constant (no predictive power), the model will still predict correctly in these regions. The authors should compare the PINN with different classical potentials that are widely accepted for AI. If they can show improved accuracy and comparable transferability under such comparisons, it would provide much stronger evidence for the advantages claimed in the paper.

Third, many recent works focus on the development of transferable machine learning force fields with different proposals. For example, Gaussian density functions (10.1021/acs.jpcc.8b08063), physics-based potentials (10.1063/1.5009502), n-body Gaussian process kernels (10.1103/PhysRevB.97.184307) just to name a very few. Many of these have a different design principle such as reducing the number of parameters, introducing physics into the models, etc. The authors should provide a more detailed literature review of the recent efforts on transferable ML force fields.

In summary, I think the authors propose an interesting solution to an important problem in ML force fields. However, in its current form the manuscript lacks key evidence needed to support their claims. If the authors are able to provide additional comparisons with (1) a more recent ML force field package and (2) a classical force field widely used for AI to support the claims of combined accuracy and transferability, I think this manuscript can be published in Nature Communications.

Response to Reviewers' comments

Re: Manuscript NCOMMS-18-27853A

“Physically-informed artificial neural networks for atomistic modeling of materials”

by G. P. Purja Pun, R. Batra, R. Ramprasad and Y. Mishin

January 21, 2019

We are grateful to all three Reviewers for providing insightful comments on our paper. This document summarizes our responses to the Reviewers and the changes made in manuscript. In particular, we have re-trained the PINN and NN potentials using the 10-fold cross-validation method; reduced the training/validation error from ~4 meV to ~ 3.5 meV without overfitting; recomputed all physical properties with the new potentials, and repeated all tests of transferability. Following requests from the Reviewers, we have constructed a stand-alone BOP potential by fitting to the same DFT database, and compared predictions of the NN, PINN and BOP potentials with those of a widely accepted EAM AI potential. We believe that these new calculations have provided additional convincing evidence of the improved transferability of the PINN model. The text of the paper has been modified to describe the new calculations, clarify some of the questions raised by the Reviewers, and add new literature citations.

Reviewer #2:

Reviewer:

I appreciate the revision, however, I find it disappointing. I feel that the authors have mainly try to argue my comments away than acting upon them properly. This is a pity as I am still not convinced. Comparison to state of the art architectures would help the paper. Clearly I would like to see the system used in an MD simulation, demonstrating the success of their approach convincingly and not alluding to its potential. Also the argument of the on the fly computations was not properly addressed. Concluding, I still cannot recommend publication of the paper in its current form.

Response:

During the previous round of revisions, we have included additional calculations and made a large number of changes to the paper, in addition to providing clarifications in response to the comments from all three Reviewers, including Reviewer #2.

Regarding the modern architectures, we have already indicated that we will consider trying convolutional networks in the future. But we cannot “act upon it” in this paper. As we explained, it is not our goal to maximize the accuracy or computational performance of the baseline model. In fact, we are moving away from the baseline model, i.e., the purely mathematical NN of any architecture, since such models are not transferable. The NN that we use in this paper for constructing the PINN potential only plays a supporting role with respect to the BOP potential. We can easily achieve the required training accuracy on the 3-4 meV/atom level using the standard feed-forward NN. This is all we need in this work. The goal of this paper is to demonstrate the principle behind the PINN model, not to optimize the computational efficiency of the training process. The latter issue may be addressed in the future.

We are not sure how we can satisfy the MD simulation request. We do have an MD code running on PINN potentials, see discussion of its performance below. The results shown in Figures S4 and S5 (liquid properties) were obtained by MD simulations. We have added a new Figure S14 demonstrating 11,000-atom MD simulations of liquid Al with the PINN potential. All properties listed in Table II were obtained by damped MD (static relaxation). Most of the DFT energies used here for training, validation and testing were obtained by ab initio MD simulations and then recomputed with the PINN and NN potentials. It is not clear to us what other MD we would need to perform to “demonstrate success” and convince the Reviewer.

The on-the-fly computations question was raised by Reviewer #3 and is discussed below.

Reviewer #3:

Reviewer:

The manuscript has been improved, some of the issues have been resolved and some relevant work was cited. The authors remark that much of the related work suggested work is not relevant due to its application in computational chemistry. Here, I strongly disagree in that the methods from the mentioned literature are very similar from a machine learning perspective, independent of their respective application.

Response:

We agree with this statement. We did add relevant literature citations during the previous round of revisions, and have added more this time. Our previous comment was prompted by the persistent requests from the Reviewers that we explore the chemical space of variables.

Reviewer:

I also disagree with the statement of the authors that on-the-fly learning would “continue forever”: e.g. using ensembles as uncertainty estimates, no further calculation will be required, once the simulation has sufficiently explored the relevant domain.

Response:

This is certainly true once we have defined the “relevant domain”. But we are facing the following problem:

The point of departure in this work is the traditional, parameter-based interatomic potentials, which presently dominate the entire field of computational materials science. As we mentioned earlier, such potentials are general-purpose type. They are not designed for one particular type of simulations - then we would indeed have a “relevant domain”. But instead, they are intended for pretty much any simulation. Once constructed, the potential goes public and is used by many people for many years for all sorts of MD and Monte Carlo simulations. Most of the atomic configurations encountered during such simulations are very different from those used during the potential development. The underlying assumption is that the potential still captures the basic physics of interatomic bonding in the material, e.g. metallic bonding in the case of EAM potentials. As such, it may not predict the accurate energies and forces, but the simulation results will still make physical sense, will still reflect the basic trends of the material’s behavior.

This is the culture in this field. Much of the accuracy is sacrificed to achieve broader applicability and enable fast computations giving access to millions of atoms and MD runs for tens of nanoseconds.

Since our goal is to develop a general-purpose potential, the concept of a “relevant domain” suggested by the Reviewer becomes very fuzzy. We never know what kind of simulations the

potential will be used for in the future. One can try to sample a very broad domain of atomic configurations by running AIMD under very diverse simulation conditions and learn these configurations on the fly. But we cannot sample the entire configuration space! This is what we meant when we said that we cannot run the on-the-fly learning forever.

The problem with the existing ML potentials is that they interpolate but cannot extrapolate. When used in extrapolation domains significantly different from the training domain, they often produce totally nonsensical results. Unfortunately, the boundary between the interpolation domain and the extrapolation domains can be very sharp (see example in Fig.4), the reason being that the extrapolation is purely mathematical. The PINN idea is to filter the ML predictions through a physics-based model such as BOP. Even if the BOP parameters predicted by the PINN potential in the extrapolation domain are inaccurate, the results will still remain physically meaningful and not simply garbage. Of course, if we keep going far away from the training domain, any model will eventually fail. But the PINN potential smears the sharp right/wrong boundary and extends the applicability of the model deeper into unexplored areas of the configuration space. In this sense, the potential does become general-purpose type, but more accurate than the traditional potentials. The transferability is achieved due to the incorporation of prior knowledge about the physics of the system, the information that this is a physical system of interacting atoms, not a string of numbers from financial markets or pixels of an image. This prior knowledge is especially important for extrapolation.

That said, we do appreciate that on-the-fly learning could be a useful strategy for potential development, including the DFT database generation and training for PINN potentials. We will explore its implementation in the future. A new reference to on-the-fly training of ML potentials has been added in the end of the fourth paragraph of Section I.

Reviewer:

My general criticism remains that PINN might obtain qualitatively "meaningful" results in the extrapolation domain, but with a seriously decreased accuracy. Since I am not too familiar with the accuracy requirements in materials science, I have to trust that this is good enough for the desired application. Beyond that, the authors have not addressed some of the raised issues sufficiently:

- Evaluation of degraded accuracy in extrapolation domain: Does PINN extrapolate better/worse than BOP? Adding BOP to some of the key figures and tables, e.g. Fig. 4 and Table I, would help to judge this.

Response:

As we understand, the Reviewer would like us to develop a stand-alone BOP potential whose 8 parameters are fitted to the DFT database and then fixed once and for all, following the traditional scheme of classical interatomic potentials. Creating such a potential was not part of our plans. However, the suggestion makes sense and we did generate a BOP potential using the same DFT database and the same software as we used for the PINN and NN potentials. The results are included in Fig.4 as suggested. As expected, the accuracy of the BOP potential is not as great as that of the PINN potential. However, it does not display the sudden switch to the unphysical wiggled behavior outside the training interval as does the NN potential.

To give the reader a more general idea about the performance of the BOP potential, we computed the BOP energies of all structures in the training and validation data set and compared them with the DFT energies. The results are shown in the new Figure 6b plotting the BOP versus DFT energies. As expected, the BOP energies are not as accurate as the PINN energies, although the general trend is captured correctly. This test shows that adjustment of the BOP parameters according to the local environment implemented in the PINN model does significantly improve the accuracy.

Reviewer:

- *Repetitions of training on different splits: the authors state in their response that they "are not able to repeat the training process multiple times with different splits". This raises serious concerns regarding the computational cost of PINN. A neural network of this size using around 3,000 training instances should usually be trained within a couple of hours. The authors should give the computation times for training and prediction of PINN.*

Response:

There are three points here that we will address in turn.

(1) One training run for a NN potential does indeed take a few hours as mentioned by the Reviewer. However, the PINN training is slower because we cannot use a well-established algorithm such as backpropagation due to the presence of the BOP layer. We thus solve the multi-dimensional unconstrained optimization problem using a Davidon-Fletcher-Powell type algorithm. The training has to be repeated multiple times starting from different initial conditions, so the training time adds up. The process is also slower because, for the model size used here, the BOP step is a factor of 2-3 slower than the NN step. Nevertheless, since the previous revision of the paper, we have been able to optimize the computations, so now the PINN training can be done in a couple of days (including multiple attempts from different initial conditions). We foresee further acceleration in the near future.

(2) Given the improved training speed, we have been able to implement the k-fold cross-validation algorithm suggested by the Reviewer (we chose k=10). The paragraph describing the training and validation processes has been rewritten accordingly. The training and validation errors reported in Table I have been obtained by averaging over the 10 rotations. Furthermore, we have been able to slightly reduce the training/validation error down to about 3.5 eV/atom without overfitting. Accordingly, all property calculations and all tests of both the PINN and NN potentials have been redone from scratch for the new versions of the potentials. Aside from some details, the conclusions of the paper remain the same. We believe that this additional work has fully addressed the concerns raised by this Reviewer.

(3) About the computational times of prediction. We are collaborating with Dr. Vesselin Yamakov at NASA Langley Research Center who has incorporated the PINN potential format in his ParaGrandMC code (<https://software.nasa.gov/software/LAR-18773-1> - the PINN-based version has not been officially released yet). ParaGrandMC is a massive parallel code for MD and Monte Carlo simulations using a number of interatomic potential forms. Here is an example of recent benchmark tests of the code:

System: Periodic simulation box of 72,000 Al atoms. 100 MD steps on 4 MPI nodes with 8 CPUs (threads) each.

Calculation of the descriptors G_i : 606.8 seconds

NN calculation: 78.4 seconds

BOP calculation: 271.8 seconds

Predictor-corrector MD algorithm: 0.1378 seconds

Run CPU time: Approx. 1023 seconds

Roughly, 10 seconds per MD step.

These numbers show that the MD speed is dominated by the calculation of the descriptors/features. Thus the additional overhead of PINN versus NN is not very significant. And of course, all these calculations are orders of magnitude faster than straight DFT calculations, which would not be even possible for the 72,000-atom system.

We would be reluctant to include these numbers in the paper because we currently work to improve the efficiency of the simulations. By the time this paper would be published, these numbers would be outdated. In fact, we are currently testing a beta-version of the code that runs on GPUs (using OpenACC), which makes it even faster.

Reviewer #4:

Reviewer:

In this work, the authors have developed a physically-informed neural network (PINN) method that uses neural networks to parameterize classical force fields. The key potential benefit of this approach is that such a combined method could have the advantage of both machine learning force fields (accuracy) and classical force fields (transferability). Transferability is an important problem for current machine learning (ML) based force fields. PINN as a general approach could be impactful if it does indeed possess the advantages claimed by the authors.

However, in reading through the paper and the other referee reports, I agree with Reviewer #2 that the manuscript lacks some key comparisons that are needed in order to make the claim of combined accuracy and transferability.

First, the baseline is weak. The authors compare PINN with a fully-connected NN with the same architecture without the BOP. The improved accuracy and transferability are not surprising because fully-connected NN are known to be prone to overfitting. The authors should compare PINN with more recent ML forces field packages (like PROPhet) and report the improvements in accuracy and transferability based on such comparisons.

Response:

(1) Overfitting was controlled by adjusting the coefficient in front of the regularization term in the objective function, see the Methods section. The NN and PINN potentials reported here are not overfitted as demonstrated by the results of the 10-fold cross-validation.

(2) The PROPhet package is based on the Behler-Parrinello symmetry functions, which are different from our local structural parameters. We do not see how a PROPhet-generated NN potential would be better than our NN potential. In addition, it would not be a baseline model for PINN due to the difference in the descriptor space.

(3) It is not our goal to improve the accuracy of training. Our NN potential is already as accurate as the DFT database (3-4 meV/atom). Its replacement by another ML potential would bring no improvements in the transferability whatsoever, because both potentials would implement purely mathematical extrapolation to unexplored domains, and thus would not be transferable.

Reviewer:

Second, the evidence of improved transferability is lacking. Transferability means that the model can extrapolate to a part of phase space not covered by the training data. The only evidence of such transferability in the manuscript is in Fig. 4, where the energy is correctly predicted outside the region of training data. However, this is in fact guaranteed by the form of BOP. This means that even if the output of the neural network is a constant (no predictive power), the model will still predict correctly in these regions.

Response:

(1) In addition to Fig.4, the evidence of transferability is presented in Figures 7, 8, S6-S13. These plots compare the DFT energies and forces with predictions of the NN and PINN potentials for atomic configurations lying outside the training/validation domain. This comparison makes it very clear that the PINN potential performs much better than the NN model.

(2) Indeed, the physically reasonable behavior in the extrapolation domain is guaranteed by the form of BOP. We agree with the Reviewer that the predictions will remain physically reasonable even if the BOP parameters are fixed. However, the predictions become more accurate if the BOP parameters are adjusted according to the local atomic environments, which is exactly what is done in the PINN model. To demonstrate this, we have constructed a stand-alone BOP potential whose parameters were fitted to the same database and then fixed. Predictions of this potential for crystal structures under strong compression are physically reasonable (by contrast to the mathematical NN model) but not as accurate as those of the PINN potential (Fig.4). The new Fig.6b demonstrates that this trend persists across the entire DFT database: the BOP potential is reasonable but not as accurate as the adaptive PINN potential. We believe that these new calculations fully address the comments from the Reviewer.

Reviewer:

The authors should compare the PINN with different classical potentials that are widely accepted for AI. If they can show improved accuracy and comparable transferability under such comparisons, it would provide much stronger evidence for the advantages claimed in the paper.

Response:

We have followed this suggestion and compared the DFT, NN and PINN energies with predictions of the widely accepted EAM AI potential [54]. The results are shown in Figures 4 and 6a. It is obvious that the EAM potential is much less accurate than both PINN and NN, especially for high-energy (far from equilibrium) structures (Fig.6a). The largest deviations arise for highly compressed DC and SC structures and uniaxial $\langle 111 \rangle$ deformation of the FCC structure. This comparison provides additional strong evidence of the advantages on the PINN model.

Reviewer:

Third, many recent works focus on the development of transferable machine learning force fields with different proposals. For example, Gaussian density functions (10.1021/acs.jpcc.8b08063), physics-based potentials (10.1063/1.5009502), n-body Gaussian process kernels (10.1103/PhysRevB.97.184307) just to name a very few. Many of these have a different design principle such as reducing the number of parameters, introducing physics into the models, etc. The authors should provide a more detailed literature review of the recent efforts on transferable ML force fields.

Response:

We have to reiterate that the existing ML force fields are not transferable. There is no reason why purely mathematical extrapolation to significantly different atomic configuration would give physically reasonable, let alone accurate, results. The transferability claims made in some of the papers are based on a different understanding of the term than our definition. We define transferability as the ability of the model to reproduce the potential energy surface, or at least its physically meaningful behavior, for atomic configurations significantly different from those used during the training. The only way to ensure this type of transferability is to incorporate prior knowledge of the basic physics of the system. In the PINN model, this is achieved by filtering the NN predictions through the BOP potential. The latter introduces the prior knowledge in the form of rather general physical properties of interatomic bonding, such as repulsion at short separations, attraction at large separations, decrease in the bond energy with the number of neighbors, bond screening by neighbors, etc. This knowledge plays especially important role in the extrapolation domains.

That said, the papers mentioned by the Reviewer are indeed very interesting. All three have been included in the bibliography. The paper by Glielmo et al [PRB 2018] introduces n-body Gaussian Process kernels, which reflect the fact that atomic interactions are multi-body type. In this sense, this does incorporate some very generic physical features, although not as explicitly as the BOP

model does. This interesting paper is mentioned in the last paragraph on page 3 and the rest of the paragraph has been slightly rephrased.

Reviewer:

In summary, I think the authors propose an interesting solution to an important problem in ML force fields. However, in its current form the manuscript lacks key evidence needed to support their claims. If the authors are able to provide additional comparisons with (1) a more recent ML force field package and (2) a classical force field widely used for AI to support the claims of combined accuracy and transferability, I think this manuscript can be published in Nature Communications.

Response:

While we do not think that using a more recent package like PROPhet would bring any additional benefits to this work (incidentally, our package is more recent than PROPhet; it is not user-oriented but its capabilities are not inferior to those of PROPhet), we have included a comparison with a widely used AI force field and demonstrated strong advantages of the proposed PINN model. Together with other additional calculations and changes in the paper, we believe that the modified version of the manuscript provides significant additional support to the claims of combined accuracy and transferability of the PINN model.

Sincerely,

G. P. Purja Pun, R. Batra, R. Ramprasad and Y. Mishin.

REVIEWERS' COMMENTS:

Reviewer #2 (Remarks to the Author):

The authors have addressed the questions I raised and added some new simulations and comparisons. I still do not feel fully convinced that the NN vs PINN comparison is showing the appropriate message and I agree with reviewer 4 that there may be some overfitting effect in the NN playing a role. So comparing here to a more recent force field package may indeed be informative. The point made in the ms is that the physics informed network makes the difference, which should be shown carefully. At the same time I also see that this comparison may not be the most crucial point in the ms given the overall nice results presented. Perhaps the authors could tone down a bit more and put things into perspective. Also, while it is true that certainly the slow computing time of the method may be a drawback now, I subscribe to the statement that this speed can be substantially increased. Still it is important to inform readers a bit more precise than now, which is trying to play down the slow computing a bit too much for my taste; this rug should be lifted.

Reviewer #3 (Remarks to the Author):

All raised issues have been addressed and I can now recommend the manuscript for publication.

Reviewer #4 (Remarks to the Author):

While the authors have considered some of my suggestions, and the addition of the BOP/EAM force field is a good improvement, I still feel that the paper should have a serious comparison with recent ML force fields. It is not surprising that a method that combines BOP and NN will perform better than both. The authors continue to claim that the only way to introduce "transferability" is through their method, but without a fair comparison with recent works it is hard to say so. In general, I like the idea presented in the paper, and the current form is good, but I am not sure it can be considered a breakthrough without comparing to a stronger ML method.

Response to Reviewers' comments

Re: Manuscript NCOMMS-18-27853-B

“Physically-informed artificial neural networks for atomistic modeling of materials”

by G. P. Purja Pun, R. Batra, R. Ramprasad and Y. Mishin

April 10, 2019

We are grateful to the Reviewers #2 and #4 for providing additional comments on our paper. We have modified the paper to address their remaining concerns. The changes made in the manuscript are highlighted in yellow.

Reviewer #2:

Reviewer:

The authors have addressed the questions I raised and added some new simulations and comparisons. I still do not feel fully convinced that the NN vs PINN comparison is showing the appropriate message and I agree with reviewer 4 that there may be some overfitting effect in the NN playing a role. So comparing here to a more recent force field package may indeed be informative. The point made in the ms is that the physics informed network makes the difference, which should be shown carefully. At the same time I also see that this comparison may not be the most crucial point in the ms given the overall nice results presented. Perhaps the authors could tone down a bit more and put things into perspective. Also, while it is true that certainly the slow computing time of the method may be a drawback now, I subscribe to the statement that this speed can be substantially increased. Still it is important to inform readers a bit more precise than now, which is trying to play down the slow computing a bit too much for my taste; this rug should be lifted

Response:

We believe that the overfitting concern has already been addressed during the previous round of revisions by using the standard cross-validation method. Table 1 shows that the training and validation errors are close to each other, which is commonly considered a demonstration that the the database was not overfitted.

Regarding a comparison with a “more recent force field package”, we are glad that the reviewer recognizes that “this comparison may not be the most crucial point in the ms given the overall nice results presented.” Nevertheless, we have followed this request and constructed another NN potential using a third-party, totally independent (from us) software package PROPNet. This packages was previously recommended by reviewer #4 as an example of a “more recent ML forces field package”. The new potential, which we call NN', shows very similar results and suffers from exactly the same problem as our NN potential: it closely follows the DFT energies within the training/validation domain and becomes totally unphysical as soon as we step outside this domain. This is illustrated by the additional curves plotted in Figure 6. The details of the NN' potential development are described in a new paragraph added on page 8 (highlighted in yellow).

We believe that this comparison provides additional demonstration of the main point of the paper: the standard (mathematical) ML potentials fail outside the training domain since the extrapolation is not guided by any physics. By contrast, the proposed physics-guided PINN potential continues to predict accurate results outside the training domain.

We thus do not see a reason to tone down our claims regarding the uniqueness of our method with respect to transferability. In fact, the comparison mentioned above only strengthens these claims.

Finally, regarding the computational speed, we are collaborating with Dr. Vesselin Yamakov at NASA Langley Research Center who has incorporated the PINN potentials in his ParaGrandMC code (<https://software.nasa.gov/software/LAR-18773-1>) and continues to make rapid progress in the code optimization. The benchmark results mentioned in our previous response to the reviewers are already outdated. Instead of being a factor of two slower than other ML force fields as was reported in the previous version of the paper, PINN is now only 25% slower. Given the rapidly growing computational power, this is a small price to pay for the greatly improved transferability. The computational aspects surrounding the PINN method will be discussed in a separate paper. In the meantime, we have modified the final paragraph of this paper to update the overhead number and add some details.

Reviewer #4:

Reviewer:

While the authors have considered some of my suggestions, and the addition of the BOP/EAM force field is a good improvement, I still feel that the paper should have a serious comparison with recent ML force fields. It is not surprising that a method that combines BOP and NN will perform better than both. The authors continue to claim that the only way to introduce "transferability" is through their method, but without a fair comparison with recent works it is hard to say so. In general, I like the idea presented in the paper, and the current form is good, but I am not sure it can be considered a breakthrough without comparing to a stronger ML method.

Response:

These comments echo reviewer #2 by recommending comparison with "with recent ML force fields". We have addressed this request by constructing a new NN potential using the recently developed PROPhet ML force field package, which was previously mentioned by this reviewer as an example of recent packages. See our response to reviewer # 2 above.

Sincerely,
G. P. Purja Pun, R. Batra, R. Ramprasad and Y. Mishin.